# PAC-Bayes Bounds for Multivariate Linear Regression and Linear Autoencoders

**Ruixin Guo**
Kent State University
rguo5@kent.edu

**Ruoming Jin**
Kent State University
rjin1@kent.edu

**Xinyu Li**
Kent State University
xli74@kent.edu

**Yang Zhou**
Auburn University
yangzhou@auburn.edu

## Abstract

Linear Autoencoders (LAEs) have shown strong performance in state-of-the-art recommender systems. However, this success remains largely empirical, with limited theoretical understanding. In this paper, we investigate the generalizability – a theoretical measure of model performance in statistical learning – of multivariate linear regression and LAEs. We first propose a PAC-Bayes bound for multivariate linear regression, extending the earlier bound for single-output linear regression by Shalaeva et al. [45], and establish sufficient conditions for its convergence. We then show that LAEs, when evaluated under a relaxed mean squared error, can be interpreted as constrained multivariate linear regression models on bounded data, to which our bound adapts. Furthermore, we develop theoretical methods to improve the computational efficiency of optimizing the LAE bound, enabling its practical evaluation on large models and real-world datasets. Experimental results demonstrate that our bound is tight and correlates well with practical ranking metrics such as Recall@K and NDCG@K.

## 1 Introduction

In recent years, simple linear recommendation models have consistently demonstrated impressive performance, often rivaling deep learning models [12, 24, 35]. In particular, linear autoencoders (LAEs) such as EASE [48] and EDLAE [49] have shown a surprising edge over classical linear methods like ALS [23]. Despite their empirical success and widespread adoption, the theoretical understanding of why LAEs perform so well remains limited. Moreover, much of recommender system research has focused heavily on empirical comparisons, where weak baselines and unreliable sampled metrics often render evaluations biased and difficult to reproduce [12, 11]. A solid theoretical foundation is therefore urgently needed to explain and justify the true performance of recommendation models beyond purely empirical assessments.

Statistical learning theory [51] provides such a foundation by estimating a model's theoretical performance over the underlying data distribution. A classic result in this area is the uniform convergence PAC bound established by Vapnik and Chervonenkis [52]. Although this bound guarantees convergence, it characterizes the *worst-case* generalization gap and is typically vacuous for large models such as neural networks, thus failing to reflect true model performance in practice [39, 13]. To address this limitation, a variant PAC framework incorporates information-theoretic techniques to bound the *expected* generalization gap, which often yields tighter results [20]. One notable example is the PAC-Bayes bound, first introduced by McAllester [37]. Recently, Dziugaite and Roy [13] empirically demonstrated that PAC-Bayes bounds can remain non-vacuous even for large neural networks, suggesting that PAC-Bayes theory provides a more accurate and practically meaningful characterization of model performance.

While statistical learning has been extensively developed for a wide range of machine learning and deep learning models [51, 6, 53], its application to recommendation systems remains largely

underexplored, with only a few exceptions [47, 46, 16]. To the best of our knowledge, it has not yet been directly applied to LAEs. In this work, we aim to advance the theoretical understanding of LAE performance through PAC-Bayes theory.

When evaluated under mean squared error, LAEs are closely related to multivariate linear regression models. Prior works have developed several PAC-Bayes bounds for linear regression. Notably, Alquier et al. [5] proposed a PAC-Bayes bound for general machine learning models with unbounded loss; Germain et al. [17] adapted this bound to linear regression models with mean squared loss, but their result does not converge; and Shalaeva et al. [45] improved Germain's bound deriving a strictly tighter and convergent version. We aim to extend existing PAC-Bayes bounds for linear regression to the setting of LAEs. Several challenges arise in doing so:

**Multivariate Data**: Existing PAC-Bayes bounds for linear regression primarily focus on the single-output setting. Extending them to the multivariate (multi-output) case is nontrivial, as the outputs can exhibit statistical dependencies. To derive a valid bound in this setting, more general assumptions on the data distribution must first be established to capture potential output dependence.

**LAE-specific Characteristics**: LAEs differ from multivariate linear regression in several important aspects. They typically operate on bounded data, so the standard Gaussian data assumption used in linear regression does not apply. Moreover, LAEs impose unique structural constraints, such as the zero-diagonal constraint on the weight matrix and the hold-out constraint on data. These characteristics must be rigorously defined and integrated into the theoretical framework.

**Computational Inefficiency**: Optimizing PAC-Bayes bounds is typically computationally expensive, as estimating the distance between prior and posterior in high-dimensional spaces is complex [41]. Since LAE models often contain hundreds of millions of parameters and are computed on large real-world datasets (Table 2), developing computationally efficient methods is crucial for practical bound evaluation.

This paper addresses the aforementioned challenges and makes the following key contributions:

- (Section 3) We generalize Shalaeva's Gaussian data assumption [45] to the multivariate setting (Assumption 3.1) and propose a corresponding PAC-Bayes bound for multivariate linear regression (Theorem 3.2), extending Shalaeva's single-output bound [45]. We further establish sufficient conditions (Theorem 3.3) that guarantee convergence for both bounds.

- (Section 4) We propose a relaxed mean squared error for evaluating LAE models and show that, under this loss, LAEs can be viewed as constrained multivariate linear regression models on bounded data. Building on this, we adapt our PAC-Bayes bound to LAEs by replacing the Gaussian data assumption with a bounded one (Assumption 4.1) and incorporating LAE-specific constraints: the zero-diagonal constraint on weights and the hold-out constraint on data (Section 4.2).

- (Section 5) We develop theoretical methods to improve the computational efficiency of optimizing the bound. Following Dziugaite and Roy [13], we restrict the prior and posterior distributions to be Gaussian (Assumption 5.1), leading to a closed-form expression for the tightest bound (Theorem 5.2). We further establish a practical upper bound with reduced complexity to mitigate the computational cost introduced by the zero-diagonal constraint (Theorem 5.4).

- (Section 6) We evaluate the bound for LAEs on real-world datasets. Experimental results demonstrate that our bound is tight and correlates well with practical evaluation metrics such as Recall@K and NDCG@K, suggesting that it effectively reflects the actual model performance of LAEs.

All proofs of the theorems, lemmas and propositions in the main paper are provided in Appendix A. Related Works are in Appendix D. Conclusions and Discussions are in Appendix E.

## 2  Preliminaries

**Notation**: We denote $S = \{(x_i, y_i)\}_{i=1}^m$ as the dataset, where $x_i \in \mathbb{R}^n$ and $y_i \in \mathbb{R}^p$ for all $i$, and $m$ is the number of samples. For each sample $(x_i, y_i)$, $x_i$ is the input and $y_i$ is the target. Let $f_W : \mathbb{R}^n \to \mathbb{R}^p$ be the linear regression model parameterized by $W \in \mathbb{R}^{p \times n}$. The model prediction is $f_W(x_i) = Wx_i$, and the mismatch between target and prediction is measured by the squared Frobenius norm loss $\|y_i - Wx_i\|_F^2$.

Denote $X = [x_1, x_2, ..., x_m] \in \mathbb{R}^{n \times m}$ as the input matrix and $Y = [y_1, y_2, ..., y_m] \in \mathbb{R}^{p \times m}$ as the target matrix. The *empirical risk*, representing the average loss on the observed dataset $S$, is then defined as $R^{\text{emp}}(W) = \frac{1}{m} \sum_{i=1}^m \|y_i - Wx_i\|_F^2 = \frac{1}{m} \|Y - WX\|_F^2$. To evaluate performance on

unseen data, we assume that each $(x_i, y_i)$ is i.i.d. sampled from an unknown distribution $\mathcal{D}$, and define the *true risk* as $R^{\text{true}}(W) = \mathbb{E}_{(x,y)\sim\mathcal{D}}\left[\|y - Wx\|_F^2\right]$.

To construct a PAC-Bayes bound, we treat $f_W$ as a stochastic model by considering $W$ a random variable. Denote $\pi$ as the prior distribution over $W$ and $\rho$ as the posterior distribution. $\pi$ represents our initial belief about $W$ before observing any data, whereas $\rho$ represents our updated belief after incorporating information from the dataset $S$ [28].

**Multivariate Linear Regression [25]**: From the definition above, the linear regression equation can be written as $Y = WX + E$, where $E = [e_1, e_2, ..., e_m] \in \mathbb{R}^{p \times m}$ is the error matrix.

Usually the first dimension of every $x_i$ is set 1 (i.e., $X_{1*}$ is a vector of all 1s) to represent the bias term. We say the linear regression is *multivariate* (or *multi-output*) if $p > 1$.

Multivariate linear regression typically assumes that the error vectors $e_i$ and $e_j$ are independent for $i \neq j$, but allows dependencies among the elements within each $e_i$. This leads to our statistical assumption stated in Assumption 3.1.

**Alquier's Bound [5] adapted to Linear Regression**: Alquier's bound is a general bound that can be applied to any model with unbounded loss. When adapted to the linear regression model using our notation, Alquier's bound can be stated as follows: Given $\pi$, for any $\lambda > 0$ and $\delta > 0$,

$$P\left(\forall\rho,\ \mathbb{E}_{W\sim\rho}[R^{\text{true}}(W)] < \mathbb{E}_{W\sim\rho}[R^{\text{emp}}(W)] + \frac{1}{\lambda}\left[D(\rho\|\pi) + \ln\frac{1}{\delta} + \Psi_{\pi,\mathcal{D}}(\lambda, m)\right]\right) \geq 1 - \delta \quad (1)$$

where $\Psi_{\pi,\mathcal{D}}(\lambda, m) = \ln\mathbb{E}_{W\sim\pi}\mathbb{E}_{S\sim\mathcal{D}^m}[e^{\lambda(R^{\text{true}}(W) - R^{\text{emp}}(W))}]$, and $D(\rho\|\pi) = \mathbb{E}_{W\sim\rho}\left[\ln\frac{\rho(dW)}{\pi(dW)}\right]$ denotes the Kullback-Leibler (KL) Divergence where the measure $\rho$ is absolutely continuous with respect to $\pi$. The loss is typically assumed following a light-tailed distribution, such as sub-Gaussian or sub-exponential, to ensure that $\Psi_{\pi,\mathcal{D}}(\lambda, m)$ is bounded [17, 18, 19].

Note that Alquier's bound holds simultaneously for all posteriors $\rho$. If replacing $\forall\rho$ with any single $\rho$ such as a Gaussian posterior or a Gibbs posterior, the bound still holds.

**Shalaeva's Bound [45]**: Shalaeva's bound is an application of Alquier's bound to the *single-output* linear regression model, i.e., the case $p = 1$. It further assumes $\mathcal{D}$ is Gaussian: Given constants $\sigma_x$ and $\sigma_e$, for any draw $(x, y) \sim \mathcal{D}$, 1. $x \sim \mathcal{N}(0, \sigma_x^2 I)$, and 2. there exists $W^* \in \mathbb{R}^{1\times n}$ such that $y = W^* x + e$, where $e \sim \mathcal{N}(0, \sigma_e^2)$ is Gaussian noise. Under this assumption, $\mathcal{D}$ is fixed in $\Psi_{\pi,\mathcal{D}}(\lambda, m)$, while $\pi$ remains unspecified. Germain et al. [17] showed that, if $\pi$ is Gaussian, $\Psi_{\pi,\mathcal{D}}(\lambda, m)$ is bounded, since the true risk $R^{\text{true}}(W)$ with $W \sim \pi$ is sub-gamma; however, their bound is independent of $m$ and does not guarantee convergence. Shalaeva et al. [45] improve this by showing that $\Psi_{\pi,\mathcal{D}}(\lambda, m)$ in fact has a strictly tighter upper bound: For any $\pi$,

$$\Psi_{\pi,\mathcal{D}}(\lambda, m) = \ln\mathbb{E}_{W\sim\pi}\frac{\exp(\lambda v_W)}{(1 + \frac{\lambda v_W}{m/2})^{m/2}} \leq \ln\mathbb{E}_{W\sim\pi}\exp\left(\frac{2\lambda^2 v_W^2}{m}\right) \quad (2)$$

where $v_W = \sigma_x^2\|W - W^*\|_2^2 + \sigma_e^2$. This bound depends on $m$ and can be used to establish convergence.

**Convergence of Shalaeva's Bound**: The convergence analysis in Shalaeva et al.'s paper [45] is presented informally. Here we formally state their results as follows: Since $\lim_{m\to\infty}(1 + \frac{\lambda v_W}{m/2})^{m/2} = \exp(\lambda v_W)$, for any $\lambda > 0$, the convergence of $\Psi_{\pi,\mathcal{D}}(\lambda, m)$ follows from

$$\lim_{m\to\infty}\Psi_{\pi,\mathcal{D}}(\lambda, m) = \lim_{m\to\infty}\ln\mathbb{E}_{W\sim\pi}\frac{\exp(\lambda v_W)}{(1 + \frac{\lambda v_W}{m/2})^{m/2}} = \ln\mathbb{E}_{W\sim\pi}\lim_{m\to\infty}\frac{\exp(\lambda v_W)}{(1 + \frac{\lambda v_W}{m/2})^{m/2}} = 0 \quad (3)$$

Upon careful examination of their analysis, we found that additional conditions are required to guarantee (3), which were not discussed in their original paper. Specifically, swapping $\lim$ and $\mathbb{E}$ is valid only under certain conditions. For example, by the *dominated convergence theorem* [44, 15], the condition can be $\mathbb{E}_{W\sim\pi}[\exp(\lambda v_W)] < \infty$. If the choice of $(\lambda, \pi)$ does not satisfy this condition, convergence is not guaranteed. These issues are discussed in Section 3.2 and Appendix B.

**Collaborative Filtering Recommenders for Implicit Feedback**: In collaborative filtering, an implicit feedback dataset is typically represented as a binary user-item interaction matrix $H \in \{0, 1\}^{n \times m}$, with $n$ items and $m$ users (Section 1.3.1.1, [1]). Each $H_{ij}$ denotes an interaction: $H_{ij} = 1$ means that user $j$ has interacted with item $i$, while $H_{ij} = 0$ means no observed interaction.

Suppose $H$ is a test set. To evaluate a model, we typically *hold out* a fraction $1 - p$ ($p \in (0, 1)$) of 1s in $H$ (Section 7.4.2, [1]; also [32, 48, 38]). Formally, we use a binary mask matrix $\boldsymbol{\Delta} \in \{0, 1\}^{n \times m}$ to perform the hold-out operation. $\boldsymbol{\Delta}$ denotes a random matrix, where each $\boldsymbol{\Delta}_{ij}$ is independently drawn from a Bernoulli distribution conditioned on $H_{ij}$: $P(\boldsymbol{\Delta}_{ij} = 1 | H_{ij} = 1) = p$, $P(\boldsymbol{\Delta}_{ij} = 0 | H_{ij} = 1) = 1 - p$ and $P(\boldsymbol{\Delta}_{ij} = 0 | H_{ij} = 0) = 1$.

Let $\Delta$ be a realization of $\boldsymbol{\Delta}$. Define the input matrix as $H^{\text{input}} = \Delta \odot H$ and the target matrix as $H^{\text{target}} = (\mathbf{1} - \Delta) \odot H$, where $\odot$ denotes the Hadamard (element-wise) product and $\mathbf{1} \in \{1\}^{n \times m}$ [1]. For any $i, j$ such that $H_{ij} = 1$, we say $H_{ij}$ is held out if $\Delta_{ij} = 0$, which yields $H_{ij}^{\text{input}} = 0$ and $H_{ij}^{\text{target}} = 1$. Consequently, $H^{\text{input}}$ retains a $p$ fraction of the 1s in $H$, while the remaining $1 - p$ fraction are held-out and moved to $H^{\text{target}}$.

A collaborative filtering model takes $H^{\text{input}}$ as input and generates a prediction matrix $H^{\text{pred}} \in \mathbb{R}^{n \times m}$. Model performance is typically evaluated by how well $(\mathbf{1} - \Delta) \odot H^{\text{pred}}$ approximates $H^{\text{target}}$. The masked prediction $(\mathbf{1} - \Delta) \odot H^{\text{pred}}$ indicates that only the entries in $H^{\text{pred}}$ that coincide with the held-out interactions contribute to the evaluation. This approximation quality is typically measured using ranking-based metrics such as Recall@K or NDCG@K (Section 7.5.3, 7.5.4, [1]).

**LAE Models and EASE [48]**: LAE models are a class of collaborative filtering models. They are typically represented by a square matrix $W \in \mathbb{R}^{n \times n}$ and trained by solving $\arg \min_W \|H - WH\|$, where $H$ denotes the training set. The model takes $H$ as input and generates a prediction $WH$, which aims to reconstruct $H$ itself. $W$ is commonly constrained by a zero diagonal (i.e., $\text{diag}(W) = 0$ [2]) to prevent overfitting towards the identity matrix $I$ [48, 49, 50]. Some studies relax this constraint by allowing a diagonal with bounded norm instead [38].

EASE is one of the most popular LAE models, obtained by solving

$$\arg \min_W \|H - WH\|_F^2 + \gamma \|W\|_F^2 \quad \text{s.t. } \text{diag}(W) = 0 \tag{4}$$

where $\gamma$ is the regularization parameter. Let $W_0$ be the solution of (4), then $W_0$ has a closed from: Let $P = \left(HH^T + \gamma I\right)^{-1}$, then $(W_0)_{ij} = 0$ if $i = j$ and $(W_0)_{ji} = -P_{ij}/P_{jj}$ if $i \neq j$.

# 3 PAC-Bayes Bound for Multivariate Linear Regression

## 3.1 The Statistical Assumption and the Bound

We first generalize Shalaeva et al.'s Gaussian data assumption [45] to the multivariate data with dependent outputs and potentially degenerate covariance. Based on this assumption, we derive our bound and show that Shalaeva's bound is a special case of ours.

**Assumption 3.1.** Let $\mu_x \in \mathbb{R}^n$, $\Sigma_x \in \mathbb{R}^{n \times n}$ be positive semi-definite, and $\Sigma_e \in \mathbb{R}^{p \times p}$ be positive-definite. Suppose $(x, y) \sim \mathcal{D}$ satisfies: 1. $x \sim \mathcal{N}(\mu_x, \Sigma_x)$; 2. there exists $W^* \in \mathbb{R}^{p \times n}$ such that $y = W^* x + e$, where $e \sim \mathcal{N}(0, \Sigma_e)$; in other words, $y|x \sim \mathcal{N}(W^* x, \Sigma_e)$.

The positive semi-definite assumption of $\Sigma_x$ allows it to be singular, implying a degenerate Gaussian distribution whose support lies on a lower-dimensional manifold embedded in $\mathbb{R}^n$. This includes the standard multivariate linear regression setting in which the first element of $x$ is 1 and the remaining $n - 1$ elements are Gaussian. In this case, the first row and first column of $\Sigma_x$ are 0.

Under Assumption 3.1, for any model $W \in \mathbb{R}^{p \times n}$, the prediction error $y - Wx = (W^* - W)x + e$ follows the Gaussian distribution $\mathcal{N}(\mu_W, \Sigma_W)$, where

$$\mu_W = \mathbb{E}[(W^* - W)x + e] = (W^* - W)\mathbb{E}[x] + \mathbb{E}[e] = (W^* - W)\mu_x$$

$$\Sigma_W = \mathbb{E}[(W^* - W)(x - \mu_x) + e)][(W^* - W)(x - \mu_x) + e]^T = (W^* - W)\Sigma_x(W^* - W)^T + \Sigma_e$$

Note that $\Sigma_W$ is positive definite due to the positive definiteness of $\Sigma_e$. Let $\Sigma_W = S^T \Lambda S$ be its eigenvalue decomposition where $S$ is orthogonal and $\Lambda = \text{diag}(\eta_1, \eta_2, ..., \eta_p)$ with $\eta_i > 0$ for all $i$. Both $S$ and $\Lambda$ depend on $W$. The **PAC-Bayes bound for multivariate linear regression** is then stated as follows:

---

[1]The same symbol $\mathbf{1}$ will be used elsewhere in this paper to represent all-ones matrices of different sizes.

[2]The notation diag is defined as follows: If $W \in \mathbb{R}^{n \times n}$, then $\text{diag}(W) \in \mathbb{R}^n$ denotes the vector consisting of the diagonal elements of $W$. If $w \in \mathbb{R}^n$, then $\text{diag}(w) \in \mathbb{R}^{n \times n}$ denotes the diagonal matrix whose diagonal entries are the elements of $w$.

**Theorem 3.2.** *Denote* $b = S\Sigma_W^{-1/2}\mu_W$. *Given* $\pi$, *for any* $\lambda > 0$ *and* $\delta > 0$,

$$P\left(\forall\rho, \mathbb{E}_{W\sim\rho}[R^{\text{true}}(W)] < \mathbb{E}_{W\sim\rho}[R^{\text{emp}}(W)] + \frac{1}{\lambda}\left[D(\rho\,\|\,\pi) + \ln\frac{1}{\delta} + \Psi_{\pi,\mathcal{D}}(\lambda, m)\right]\right) \geq 1 - \delta \quad (5)$$

*where*

$$\Psi_{\pi,\mathcal{D}}(\lambda, m) = \ln\mathbb{E}_{W\sim\pi}\left[\exp\left(\lambda\left(\text{tr}(\Sigma_W) + \mu_W^T\mu_W\right)\right)\frac{\exp\left(\sum_{i=1}^{p}\frac{-\lambda m b_i^2\eta_i}{m+2\lambda\eta_i}\right)}{\prod_{i=1}^{p}\left(1 + 2\lambda\eta_i/m\right)^{m/2}}\right] \leq \ln\mathbb{E}_{W\sim\pi}\exp\left(\frac{2\lambda^2\|\Sigma_W\|_F^2}{m}\right)$$

The bound of Theorem 3.2 is a general case of Shalaeva's bound. It can be reduced to Shalaeva's bound by taking $p = 1$, $\mu_x = 0$, $\Sigma_x = \sigma_x^2 I$ and $\Sigma_e = \sigma_e^2$ for some constants $\sigma_x, \sigma_e$.

### 3.2 Convergence Analysis

This section presents the convergence analysis of Theorem 3.2. We provide a sufficient condition based on the *dominated convergence theorem* that guarantees convergence, thereby completing and rigorously formalizing the convergence analysis of Shalaeva's bound [45]. This condition is stated as follows:

**Theorem 3.3.** *If* $\lambda$ *and* $\pi$ *satisfies* $\mathbb{E}_{W\sim\pi}\left[\exp\left(\lambda\|(\Sigma_x + \mu_x\mu_x^T)^{1/2}(W^* - W)\|_F^2\right)\right] < \infty$, *then* $\lim_{m\to\infty}\Psi_{\pi,\mathcal{D}}(\lambda, m) = 0$.

Here are some examples of the combinations $(\lambda, \pi)$ that satisfy the condition of Theorem 3.3:

**Example 3.4.** Let $\pi$ be a distribution with bounded support, then for any $\lambda > 0$, the condition holds, because there exists a constant $G > 0$ with $\|W\|_F < G$ such that

$$\mathbb{E}_{W\sim\pi}\left[\exp\left(\lambda\|(\Sigma_x + \mu_x\mu_x^T)^{1/2}(W^* - W)\|_F^2\right)\right] \leq \mathbb{E}_{W\sim\pi}\left[\exp\left(\lambda\|(\Sigma_x + \mu_x\mu_x^T)^{1/2}\|_F^2\|W^* - W\|_F^2\right)\right]$$

$$\leq \mathbb{E}_{W\sim\pi}\left[\exp\left(\lambda\|(\Sigma_x + \mu_x\mu_x^T)^{1/2}\|_F^2\left(\|W^*\|_F + \|W\|_F\right)^2\right)\right] < \exp\left(\lambda\|(\Sigma_x + \mu_x\mu_x^T)^{1/2}\|_F^2\left(\|W^*\|_F + G\right)^2\right) < \infty$$

**Example 3.5.** Let $\pi$ be a Gaussian distribution parameterized by $\mathcal{U}_0 \in \mathbb{R}^{n\times n}$ and $\sigma > 0$, such that each $W_{ij}$ is independently drawn from $\mathcal{N}((\mathcal{U}_0)_{ij}, \sigma^2)$. Let $\Sigma_x + \mu_x\mu_x^T = Q^T\Lambda Q$ be its eigenvalue decomposition, where $\Lambda = \text{diag}(\nu_1, \nu_2, ..., \nu_n)$ and $\nu_1$ is the largest eigenvalue, then

$$\mathbb{E}_{W\sim\pi}\left[\exp\left(\lambda\|(\Sigma_x + \mu_x\mu_x^T)^{1/2}(W^* - W)\|_F^2\right)\right] = \prod_{i=1}^{p}\prod_{j=1}^{p}\frac{\exp\left(\frac{\lambda\nu_j\left(Q_{j*}(W^* - \mathcal{U}_0)_{*i}\right)^2}{1 - 2\lambda\sigma^2\nu_j}\right)}{\left(1 - 2\lambda\sigma^2\nu_j\right)^{1/2}}$$

In this case, the sufficient condition holds for any $\lambda \in (0, \frac{1}{2\nu_1\sigma^2})$.

Applying Theorem 3.3 to Shalaeva's bound, then (3) is guaranteed if $\lambda$ and $\pi$ satisfies $\mathbb{E}_{W\sim\pi}\left[\exp\left(\lambda\sigma_x^2\|W^* - W\|_2^2\right)\right] < \infty$. Moreover, since

$$\mathbb{E}_{W\sim\pi}\left[\exp\left(\lambda\sigma_x^2\|W^* - W\|_2^2\right)\right] < \mathbb{E}_{W\sim\pi}\left[\exp\left(\lambda\sigma_x^2\|W^* - W\|_2^2 + \lambda\sigma_e^2\right)\right] = \mathbb{E}_{W\sim\pi}[\exp(\lambda v_W)]$$

a sufficient condition is therefore $\mathbb{E}_{W\sim\pi}[\exp(\lambda v_W)] < \infty$.

## 4 PAC-Bayes Bound for LAEs

This section presents a PAC-Bayes bound for LAEs. The model $W$ can be obtained using any training method, such as EASE, EDLAE or ELSA; our bound only focuses on analyzing its test performance and is independent of the training procedure.

### 4.1 Adapting to Bounded Data Assumption

Most real-world recommendation datasets are bounded rather than Gaussian. For example, the user-item interaction matrix $H$ introduced in Section 2 is binary. If we assume that each user vector $H_{*i}$ is i.i.d. sampled from an underlying distribution, this distribution must have bounded support. Therefore, to adapt the PAC-Bayes bound for multivariate linear regression to recommendation datasets, we replace the Gaussian assumption on $\mathcal{D}$ (Assumption 3.1) with the following bounded-support assumption:

**Assumption 4.1.** Suppose $\mathcal{D}$ is characterized by three finite cross-correlation matrices $\Sigma_{xx} = \mathbb{E}_{(x,y)\sim\mathcal{D}}[xx^T]$, $\Sigma_{xy} = \mathbb{E}_{(x,y)\sim\mathcal{D}}[xy^T]$ and $\Sigma_{yy} = \mathbb{E}_{(x,y)\sim\mathcal{D}}[yy^T]$, where $\Sigma_{xx}$ is positive definite.

This assumption is indeed general. It holds for all $\mathcal{D}$ with bounded support, and also holds for certain $\mathcal{D}$ with unbounded support such as Gaussian, since Assumption 3.1 implies Assumption 4.1. Consequently, it enables the derivation of a more general bound, although in this work we focus on its application to bounded-data settings.

We first note that the true risk under Assumption 4.1 can be expressed in an explicit form:

**Lemma 4.2.** *Given any $W$, the true risk can be expressed as*

$$R^{\text{true}}(W) = ||W\Sigma_{xx}^{1/2} - \Sigma_{xy}^T\Sigma_{xx}^{-1/2}||_F^2 - ||\Sigma_{xy}^T\Sigma_{xx}^{-1/2}||_F^2 + \text{tr}(\Sigma_{yy}) \tag{6}$$

Then, in (1), we have the upper bound $\Psi_{\pi,\mathcal{D}}(\lambda, m) \leq \ln \mathbb{E}_\pi\left[e^{\lambda R^{\text{true}}(W)}\right]$, which is obtained by removing $-R^{\text{emp}}(W)$ due to its non-positivity. This upper bound, originally used by Germain et al. (Appendix A.4, [17]), does not ensure convergence since it is independent of $m$, but it simplifies computation. By plugging in (6), we get

**Proposition 4.3.** *Denote $B = -\Sigma_{xy}^T\Sigma_{xx}^{-1/2}$ and $C = e^{\lambda\left(\text{tr}(\Sigma_{yy}) - ||\Sigma_{xy}^T\Sigma_{xx}^{-1/2}||_F^2\right)}$. Then (1) holds, with $\Psi_{\pi,\mathcal{D}}(\lambda, m)$ upper-bounded by*

$$\Psi_{\pi,\mathcal{D}}(\lambda, m) \leq \ln \mathbb{E}_\pi\left[e^{\lambda R^{\text{true}}(W)}\right] = \ln C\, \mathbb{E}_\pi\left[e^{\lambda||W\Sigma_{xx}^{1/2} + B||_F^2}\right] \tag{7}$$

### 4.2 Applying the PAC-Bayes Bound to LAEs

Recall from Section 2 that the LAE model is defined by a squared matrix $W \in \mathbb{R}^{n\times n}$, and evaluated by comparing the closeness between the masked prediction $(\mathbf{1} - \Delta) \odot (W(\Delta \odot H))$ and the target $(\mathbf{1} - \Delta) \odot H$. This closeness is typically measured by ranking-based metrics such as Recall@K and NDCG@K, which are discrete and difficult to analyze statistically. To simplify the analysis, we use *mean squared error* (MSE; see Section 7.5.1, [1]) instead. The classic MSE is defined as the mean squared Frobenius norm of all held-out interactions:

$$\frac{1}{m}||(\mathbf{1} - \Delta) \odot H - (\mathbf{1} - \Delta) \odot (W(\Delta \odot H))||_F^2$$

where the mask $1 - \Delta$ on prediction distinguishes it from a multivariate linear regression. We relax the MSE by removing this mask, allowing all predicted interactions to participate in the evaluation rather than only the held-out ones:

$$\frac{1}{m}||(\mathbf{1} - \Delta) \odot H - W(\Delta \odot H)||_F^2 \tag{8}$$

Let $X = \Delta \odot H$ be the input and $Y = (\mathbf{1} - \Delta) \odot H$ be the target. The 1s in $X_{*j}$ represent items observed by user $j$, while the 1s in $Y_{*j}$ represent items that are hidden but potentially of interest to the user. The classic MSE only evaluates on held-out items, i.e., those with $(X_{ij} = 0, Y_{ij} = 1)$. In contrast, our relaxed MSE (8) also accounts for items with with $(X_{ij} = 0, Y_{ij} = 0)$, indicating that items unobserved and unlikely to interest the user should not be recommended; and those with $(X_{ij} = 1, Y_{ij} = 0)$, indicating that the model should avoid recommending items already observed by the user (see Section 7.3.4, [1]). Consequently, (8) can be viewed as a special case of the empirical risk of linear regression $R^{\text{emp}}(W) = \frac{1}{m}||Y - WX||_F^2$, with output dimension $p = n$, and under the following **LAE-specific constraints**:

1. Hold-out constraint on $X$ and $Y$: For any $i, j$, $X_{ij}$ and $Y_{ij}$ are either 0 or 1, but cannot both be 1.

2. Zero-diagonal constraint on $W$: $\text{diag}(W) = 0$ (Optional).

Since both $X$ and $Y$ are derived from $H$ and $\Delta$, the true risk can be defined by introducing statistical assumptions on $H$ and $\Delta$ respectively. For $H$, we assume that each user vector $H_{*j}$ is i.i.d. sampled from a multivariate Bernoulli distribution $\mathcal{M}$, and denote $\Sigma_{hh} = \mathbb{E}_{h\sim\mathcal{M}}[hh^T]$ as the cross-correlation matrix of $\mathcal{M}$. For $\Delta$, denote $\mathcal{B}$ as the distribution from which each column $\Delta_{*j}$ is independently drawn. Note that $\mathcal{B}$ depends on $\mathcal{M}$, and this dependence encodes the hold-out mechanism: for $\boldsymbol{\delta} \sim \mathcal{B}$ and $h \sim \mathcal{M}$, $P(\boldsymbol{\delta}_i = 1|h_i = 1) = p$, $P(\boldsymbol{\delta}_i = 0|h_i = 1) = 1 - p$ and $P(\boldsymbol{\delta}_i = 0|h_i = 0) = 1$. In the true risk $R^{\text{true}}(W) = \mathbb{E}_{(x,y)\sim\mathcal{D}}\left[||y - Wx||_F^2\right]$, by plugging in $x = \boldsymbol{\delta} \odot h$, $y = (\mathbf{1} - \boldsymbol{\delta}) \odot h$, we get

$$R^{\text{true}}(W) = \mathbb{E}_{\boldsymbol{\delta} \sim \mathcal{B}, h \sim \mathcal{M}} \left[ ||(\mathbf{1} - \boldsymbol{\delta}) \odot h - W(\boldsymbol{\delta} \odot h)||_F^2 \right] \tag{9}$$

By Lemma 4.2, we further obtain the following result:

**Lemma 4.4.** *(9) can be written in the same form as (6) by plugging in*

$$\Sigma_{xx} = p^2 \Sigma_{hh} + p(1-p)(I \odot \Sigma_{hh}), \ \Sigma_{yy} = (1-p)^2 \Sigma_{hh} + p(1-p)(I \odot \Sigma_{hh}), \ \Sigma_{xy} = p(1-p)(\Sigma_{hh} - I \odot \Sigma_{hh})$$

*Furthermore, $\Sigma_{xx}$ is positive definite if $\Sigma_{hh}$ is positive definite.*

If $W$ is subject to a zero-diagonal constraint, such as models trained from EASE, EDLAE or ELSA, then an additional condition $\text{diag}(W) = 0$ is applied to both the empirical risk (8) and the true risk (9). The **PAC-Bayes bound for LAEs** is formed by plugging (8) and (9) into (1). Since Assumption 4.1 holds for this bound, it directly leads to (7).

## 5 Practical Methods for Computing the PAC-Bayes Bound for LAEs

In the PAC-Bayes bound for LAEs proposed in Section 4, the choice of $\pi$ and $\rho$ is so far unspecified, and the computation of the bound has not yet been addressed. This section develops theoretical methods to improve the computational efficiency, enabling the bound to be evaluated on large models and datasets.

Our goal is to optimize the right hand side of (1): Given $\delta$, find $\pi, \rho, \lambda$ that minimize

$$\underbrace{\mathbb{E}_{W \sim \rho}[R^{\text{emp}}(W)] + \frac{1}{\lambda} D(\rho \,||\, \pi)}_{\text{part 1}} + \frac{1}{\lambda} \ln \frac{1}{\delta} + \frac{1}{\lambda} \underbrace{\Psi_{\pi, \mathcal{D}}(\lambda, m)}_{\text{part 2}} \tag{10}$$

with $R^{\text{emp}}(W)$ given by (8) and $R^{\text{true}}(W)$ given by (9). However, solving for $\lambda, \pi, \rho$ simultaneously is generally intractable [4]. We therefore consider a weaker problem: Given $\lambda$ and $\pi$, we optimize (10) with respect to $\rho$. We compute the optimal $\rho$ for different choices of $\lambda$ and $\pi$, and select the combination yielding the tightest bound.

We discuss the computation of part 1 of (10) in Section 5.1 and part 2 in Section 5.2.

### 5.1 Closed-form Solution for the Optimal $\rho$ under Gaussian Constraint

Given $\pi$ and $\lambda$, we search for the optimal $\rho$ by

$$\arg\min_{\rho} \mathbb{E}_{W \sim \rho}[R^{\text{emp}}(W)] + \frac{1}{\lambda} D(\rho \,||\, \pi) \tag{11}$$

Not all choices of $\pi$ and $\rho$ make (11) easy to solve. For example, given any $\pi$, the optimal $\rho$ is the Gibbs posterior, defined as $\rho(dW) = \frac{e^{-\lambda R^{\text{emp}}(W)} \pi(dW)}{\mathbb{E}_{\pi}[e^{-\lambda R^{\text{emp}}(W)}]}$ [4]. However, the Gibbs posterior is generally a complex distribution without a closed-form density function or parameterization, making it analytically intractable in practice.

To obtain a tractable and efficient solution, we instead optimize (11) under the constraint that $\pi, \rho$ are restricted to specific distribution families. Notably, Dziugaite and Roy [13] proposed a practical way to compute PAC-Bayes bounds for deep neural networks by assuming $\pi$ and $\rho$ to be entry-wise Gaussian distributions, which allows the KL-Divergence $D(\rho \,||\, \pi)$ to be computed analytically. We follow Dziugaite and Roy's assumption [13] and formally state it as follows:

**Assumption 5.1.** Let $\mathcal{A}, \mathcal{B} \in \mathbb{R}^{n \times n}$ with $\mathcal{B} \geq 0$ (entry-wise non-negative), and denote $\bar{\mathcal{N}}(\mathcal{A}, \mathcal{B})$ as the entry-wise Gaussian distribution such that $W \sim \bar{\mathcal{N}}(\mathcal{A}, \mathcal{B})$ means each $W_{ij}$ is independently drawn from $\mathcal{N}(\mathcal{A}_{ij}, \mathcal{B}_{ij})$. Assume $\rho = \bar{\mathcal{N}}(\mathcal{U}, \mathcal{S})$ and $\pi = \bar{\mathcal{N}}(\mathcal{U}_0, \sigma^2 J)$, where $\mathcal{U}, \mathcal{U}_0, \mathcal{S} \in \mathbb{R}^{n \times n}$ with $\mathcal{S} > 0$ (entry-wise positive), $J \in \{1\}^{n \times n}$ is the all-ones matrix, and $\sigma > 0$.

Applying the constraint $\text{diag}(W) = 0$ to $\rho$ and $\pi$ implies setting $\text{diag}(\mathcal{U}) = 0, \text{diag}(\mathcal{S}) = 0$, $\text{diag}(\mathcal{U}_0) = 0$ and $\text{diag}(\sigma^2 J) = 0$, since a deterministically zero random variable has zero mean and zero variance.

Since neural network models are typically non-linear and do not admit closed-form solutions for the optimal $\rho$, Dziugaite and Roy [13] solved for the optimal $\rho$ using stochastic gradient descent

with gradients estimated via Monte Carlo sampling, which requires a trade-off between sample size and computational cost. Surprisingly, due to the linearity of to LAE models, we find that their assumption adapted to LAEs admits a closed-form solution for the optimal $\rho$, as shown in Theorem 5.2. This closed-form solution enables efficient, direct computation of $\rho$, avoiding sampling, iteration or trade-off procedures.

**Theorem 5.2.** *(a) Under Assumption 5.1, the closed-form solution of the optimal $\rho$ of (11) is given by*

$$\mathcal{U} = \left( \frac{1}{m} Y X^T + \frac{1}{2\lambda\sigma^2} \mathcal{U}_0 \right) \left( \frac{1}{m} X X^T + \frac{1}{2\lambda\sigma^2} I \right)^{-1}, \ \mathcal{S}_{ij} = \frac{1}{\frac{2\lambda}{m} X_{j*} X_{j*}^T + \frac{1}{\sigma^2}} \ \ for \ i,j \in \{1,2,...,n\}$$

*(b) If we add the constraint $\mathrm{diag}(W) = 0$ to both $\rho$ and $\pi$, then the optimal $\rho$ becomes*

$$\mathcal{S}_{ij} = \frac{1}{\frac{2\lambda}{m} X_{j*} X_{j*}^T + \frac{1}{\sigma^2}}, \ \mathcal{S}_{ii} = 0 \ \ for \ i,j \in \{1,2,...,n\} \ and \ i \neq j$$

$$\mathcal{U} = \left( \frac{1}{m} Y X^T + \frac{1}{2\lambda\sigma^2} \mathcal{U}_0 - \frac{1}{2} \mathrm{diag}(x) \right) \left( \frac{1}{m} X X^T + \frac{1}{2\lambda\sigma^2} I \right)^{-1}$$

*where*

$$x = 2 \cdot \mathrm{diag}\left[ \left( \frac{1}{m} Y X^T + \frac{1}{2\lambda\sigma^2} \mathcal{U}_0 \right) \left( \frac{1}{m} X X^T + \frac{1}{2\lambda\sigma^2} I \right)^{-1} \right] \oslash \mathrm{diag}\left[ \left( \frac{1}{m} X X^T + \frac{1}{2\lambda\sigma^2} I \right)^{-1} \right]$$

*and $\oslash$ denotes element-wise division.*

## 5.2 Reducing the Complexity of $\Psi_{\pi,\mathcal{D}}(\lambda, m)$ under the Zero-Diagonal Constraint

Under Assumptions 4.1 and 5.1, the closed-form expression of $\Psi_{\pi,\mathcal{D}}(\lambda, m)$ is too complex to derive explicitly, making direct computation infeasible. We therefore compute the upper bound given in (7) instead. This section shows that, under the zero-diagonal constraint, this computation has a high complexity of $O(n^4)$, which we reduce to $O(n^3)$ by establishing a simpler upper bound.

We first consider the case without the constraint $\mathrm{diag}(W) = 0$. Since we assume $\pi = \bar{\mathcal{N}}(\mathcal{U}_0, \sigma^2 J)$ in Assumption 5.1, $W_{i*}^T \sim \mathcal{N}((\mathcal{U}_0)_{i*}^T, \sigma^2 I)$, thus $(W_{i*}\Sigma_{xx}^{1/2} + B_{i*})^T = \Sigma_{xx}^{1/2} W_{i*}^T + B_{i*}^T \sim \mathcal{N}(\Sigma_{xx}^{1/2}(\mathcal{U}_0)_{i*}^T + B_{i*}^T, \sigma^2 \Sigma_{xx})$. In this case, the $\mathbb{E}_\pi\left[ e^{\lambda R^{\mathrm{true}}(W)} \right]$ term in (7) can be further expressed as follows:

**Proposition 5.3.** *Let $A = \sigma^2 \Sigma_{xx}$, and $A = S^T \Lambda S$ be the eigenvalue decomposition where $S$ is orthogonal and $\Lambda = \mathrm{diag}(\eta_1, \eta_2, ..., \eta_n)$ with $\eta_1 \geq ... \geq \eta_n \geq 0$ [3]. Denote $\mu^i = \Sigma_{xx}^{1/2}(\mathcal{U}_0)_{i*}^T + B_{i*}^T$.*

$$\mathbb{E}_\pi\left[ e^{\lambda R^{\mathrm{true}}(W)} \right] = C \prod_{i=1}^n \prod_{j=1}^n \frac{\exp\left( \frac{\lambda(\bar{b}_j^i)^2 \eta_j}{1 - 2\lambda\eta_j} \right)}{\left( 1 - 2\lambda\eta_j \right)^{1/2}}, \quad \text{where } \bar{b}^i = S A^{-1/2} \mu^i \tag{12}$$

The computational complexity of (12) is $O(n^3)$, mainly due to the eigenvalue decomposition of $A$.

Now we discuss the case that $\mathrm{diag}(W) = 0$ is applied. Denote $\pi'$ as the distribution $\pi$ with the constraint $\mathrm{diag}(W) = 0$, that is, for $W \sim \pi'$, $W_{ii} = 0$ for all $i$. Then $\pi' = \bar{\mathcal{N}}(\mathcal{U}_0, \sigma^2(J - I))$ where $\mathrm{diag}(\mathcal{U}_0) = 0$, and $W_{i*}^T \sim \mathcal{N}\left( (\mathcal{U}_0)_{i*}^T, \sigma^2(I - I^i) \right)$ where $I^i$ is a matrix with $I_{ii}^i = 1$ and other entries being 0. Therefore, $(W_{i*}\Sigma_{xx}^{1/2} + B_{i*})^T \sim \mathcal{N}\left( \Sigma_{xx}^{1/2}(\mathcal{U}_0)_{i*}^T + B_{i*}^T, \sigma^2(\Sigma_{xx} - (\Sigma_{xx}^{1/2})_{*i}(\Sigma_{xx}^{1/2})_{*i}^T) \right)$. Denote $A^{(i)} = \sigma^2(\Sigma_{xx} - (\Sigma_{xx}^{1/2})_{*i}(\Sigma_{xx}^{1/2})_{*i}^T)$, then $A^{(i)}$ is singular and positive semi-definite. Let $A^{(i)} = S^{(i)T} \Lambda^{(i)} S^{(i)}$ be the eigenvalue decomposition where $S^{(i)}$ is orthogonal and $\Lambda^{(i)} = \mathrm{diag}(\eta_1^{(i)}, \eta_2^{(i)}, ..., \eta_n^{(i)})$ with $\eta_1^{(i)} \geq ... \geq \eta_n^{(i)} \geq 0$. Then

$$\mathbb{E}_{\pi'}\left[ e^{\lambda R^{\mathrm{true}}(W)} \right] = C \prod_{i=1}^n \prod_{j=1}^n \frac{\exp\left( \frac{\lambda(b_j^{(i)})^2 \eta_j^{(i)}}{1 - 2\lambda\eta_j^{(i)}} \right)}{\left( 1 - 2\lambda\eta_j^{(i)} \right)^{1/2}}, \quad \text{where } b^{(i)} = S^{(i)}(A^{(i)})^{-1/2} \mu^i \tag{13}$$

---

[3] We slightly abuse the notation of $S$ and $\Lambda$. In Theorem 3.2, they denote the decomposition of $\Sigma_W$, whereas here they denote the decomposition of $A$.

The issue with (13) is its high computational complexity: We need to compute the eigenvalue decomposition for each $A^{(i)}$ in order to obtain $S^{(i)}$ and $\Lambda^{(i)}$. Since each eigenvalue decomposition costs $O(n^3)$, the computation of (13) costs $O(n^4)$, which is impractical for large $n$.

Since the direct computation of $\mathbb{E}_{\pi'}\left[e^{\lambda R^{\mathrm{true}}(W)}\right]$ is difficult, we instead compute an upper bound, as established by the following theorem:

**Theorem 5.4.** *Given (12) and (13), for any $\lambda \in \left(0, \frac{1}{2\eta_1}\right)$ where $\eta_1$ is the largest eigenvalue of $A$,*

$$\mathbb{E}_{\pi'}\left[e^{\lambda R^{\mathrm{true}}(W)}\right] \leq \mathbb{E}_{\pi}\left[e^{\lambda R^{\mathrm{true}}(W)}\right]$$

Theorem 5.4 holds for any $\mathcal{U}_0$, including the special case where $\mathrm{diag}(\mathcal{U}_0) = 0$ for both $\pi'$ and $\pi$. This theorem allows us to compute $\mathbb{E}_{\pi}\left[e^{\lambda R^{\mathrm{true}}(W)}\right]$ instead of $\mathbb{E}_{\pi'}\left[e^{\lambda R^{\mathrm{true}}(W)}\right]$, thereby reducing the complexity from $O(n^4)$ to $O(n^3)$.

### 5.3 The Final Bound and the Algorithm for its Computation

The final step in computing the bound is to select $\lambda$ that yields the tightest bound. Following [4], we search over a finite grid $\Lambda = \{\lambda_1, \lambda_2, ..., \lambda_L\}$ where $L$ is the number of elements in $\Lambda$ and $\lambda_i > 0$ for $i \in \{1, 2, ..., L\}$; details are provided in Appendix C. Applying this grid search to (1), we obtain the final bound: Given $\pi$, for any $\lambda \in \Lambda$ and $\delta > 0$, with probability at least $1 - \delta$, the following bound holds for any $\rho$:

$$\mathbb{E}_{W \sim \rho}[R^{\mathrm{true}}(W)] \leq \mathbb{E}_{W \sim \rho}[R^{\mathrm{emp}}(W)] + \frac{1}{\lambda}\left[D(\rho \,\|\, \pi) + \ln\frac{L}{\delta} + \ln\mathbb{E}_{\pi}\left[e^{\lambda R^{\mathrm{true}}(W)}\right]\right] \tag{14}$$

We now summarize computation of (14) under the LAE setting in Algorithm 1. By default, the algorithm assumes that the zero-diagonal constraint on $W$ is applied. For unconstrained $W$, the algorithm can be adapted by switching the solution for $\rho$ from Theorem 5.2 (b) to Theorem 5.2 (a), and by computing $D(\rho \,\|\, \pi)$ using the unconstrained case (25) instead of (42).

Note that this algorithm requires $\Sigma_{hh} = \mathbb{E}_{h \sim \mathcal{M}}[hh^T]$ as input, which depends on $\mathcal{M}$. In practice, $\mathcal{M}$ may be an oracle distribution, making $\Sigma_{hh}$ unknown and inaccessible. However, there are practical scenarios where a non-oracle $\mathcal{M}$ is available. For example, if there exists a larger and fixed dataset $H^{\mathrm{whole}} \in \{0, 1\}^{n \times m'}$ $(m' > m)$ such that the columns of $H$ are sampled without replacement from the columns of $H^{\mathrm{whole}}$, we can take $\mathcal{M}$ as the *population distribution* over the columns of $H^{\mathrm{whole}}$. This is similar to the matrix completion setting of Srebro et al. [47], where all entries of a fixed matrix are treated as underground truth and observed entries are sampled from them. Under this assumption, $\Sigma_{hh} = \frac{1}{m'}H^{\mathrm{whole}}(H^{\mathrm{whole}})^T$ is known and accessible. We provide further discussion in Appendix E.1.

---

**Algorithm 1** Computing the PAC-Bayes bound for LAEs

---

**Input:** $\Sigma_{hh}, p, \delta, \sigma, \Lambda = \{\lambda_1, \lambda_2, ..., \lambda_L\}, X, Y$, and an LAE model $W$ (with $\mathrm{diag}(W) = 0$).
Compute $\Sigma_{xx}, \Sigma_{xy}, \Sigma_{yy}$ with $\Sigma_{hh}, p$ by Lemma 4.4.
Set $\pi = \mathcal{N}(W, \sigma^2 I)$ (i.e., let $\mathcal{U}_0 = W$ such that $W$ is the mean prior of $\pi$).
Let $G = \{\}$ be a set to store the results.
**for** each $\lambda_i$ **in** $\Lambda$:
    Compute $\rho = \bar{\mathcal{N}}(\mathcal{U}, \mathcal{S})$ with $\pi, \lambda_i$ by Theorem 5.2 (b).
    Compute $D(\rho \,\|\, \pi)$ with $\rho, \pi$ by (42) in Appendix F.
    Compute $\mathbb{E}_{W \sim \rho}[R^{\mathrm{emp}}(W)]$ with $\rho, X, Y$ by (40) in Appendix F.
    Compute $\mathbb{E}_{W \sim \rho}[R^{\mathrm{true}}(W)]$ with $\rho, \Sigma_{xx}, \Sigma_{xy}, \Sigma_{yy}$ by (41) in Appendix F, and let it be the left hand side of (14), denoted as $\mathrm{LH}_i$.
    Compute $\mathbb{E}_{\pi}\left[e^{\lambda R^{\mathrm{true}}(W)}\right]$ with $\pi, \Sigma_{xx}, \Sigma_{xy}, \Sigma_{yy}, \lambda_i$ by (12).
    Compute the right hand side of (14), denoted as $\mathrm{RH}_i$, with $\mathbb{E}_{W \sim \rho}[R^{\mathrm{emp}}(W)], D(\rho \,\|\, \pi), \mathbb{E}_{\pi}\left[e^{\lambda R^{\mathrm{true}}(W)}\right]$.
    Append $(\mathrm{LH}_i, \mathrm{RH}_i)$ to $G$.
**Output:** the pair $(\mathrm{LH}^*, \mathrm{RH}^*)$ in $G$, where $\mathrm{RH}^* = \min_{1 \leq i \leq L}\{\mathrm{RH}_i\}$.

---

# 6 Experiments

In this section, we conduct experiments to compute the PAC-Bayes bound for LAEs using Algorithm 1 on real-world datasets, evaluate the tightness of the bound, and empirically assess its correlation with practical ranking metrics such as Recall@K and NDCG@K.

We adopt the *strong generalization* evaluation setting, which divides the entire dataset into a training set and a test set with disjoint users [48, 38]. Let the entire dataset be $H^{\text{whole}} \in \{0,1\}^{n \times m'}$. We split it into a training set $H^{\text{train}} \in \{0,1\}^{n \times (m'-m)}$ and a test set $H^{\text{test}} \in \{0,1\}^{n \times m}$ by setting $m = 0.3m'$. The test set $H^{\text{test}}$ is further split into an input matrix $X$ and a target matrix $Y$, with a hold-out fraction $1 - p = \frac{1}{2}$. The LAE model $W$ is obtained by solving the EASE objective (4) on the training set $H^{\text{train}}$ (The EASE model can also be replaced by other LAE models such as EDLAE or ELSA. Our bound only focuses on evaluation and is independent of the training method.). We set $\gamma$ in (4) to values of $50, 100, 200, 500, 1000, 2000$ and $5000$ to generate seven different LAE models and evaluate them accordingly. Other inputs of the algorithm are set as follows: $\delta = 0.01$, $\sigma = 0.001$, $\Lambda = \{1, 2, 4, 8, 16, 32, 64, 128, 256, 512\}$.

Our experiments run on a machine with 500 GB RAM and an Nvidia A100 GPU. The GPU has 80 GB RAM. We use three datasets: MovieLens 20M (ML 20M), Netflix and MSD, with their details shown in Table 2 in Appendix F.

The results are presented in Table 1. On the left side, each pair (LH, RH) is the output of Algorithm 1, where LH and RH represent the left-hand side and right-hand side of (14), respectively. Detailed values of the components of RH are provided in Table 3 in Appendix F. The results demonstrate that our bound is tight: in all cases, RH is within 3 times LH, in contrast to Dziugaite and Roy's non-vacuous bound for deep neural networks [13], where RH can reach up to 10 times LH in their experiments.

The right side of Table 1 reports Recall@50 and NDCG@100 for each model on test sets, with both metrics referenced from the EASE paper [48]. Across all datasets, models with smaller LH and RH generally achieve higher Recall@50 and NDCG@100. This negative correlation reflects the expected relationship: LH and RH are loss-based, where lower values are better; Recall@50 and NDCG@100 are ranking metrics, where higher values are better. Although minor deviations exist – for example, on MSD, the best LH/RH occur at $\gamma = 1000$, while the best Recall@50/NDCG@100 occur at $\gamma = 500$ – the overall trend demonstrates a strong alignment between LH/RH and Recall@50/NDCG@100, suggesting that our bound effectively reflects the practical performance of LAE models. Further discussion on this alignment is provided in Appendix E.4.

Table 1: Experiment results

| Models | | PAC-Bayes Bound for LAEs | | | | Ranking Performance | | |
|---|---|---|---|---|---|---|---|---|
| | | ML 20M | Netflix | MSD | | ML 20M | Netflix | MSD |
| $\gamma = 50$ | LH | 61.66 | 87.22 | 15.96 | Recall@50 | 0.3434 | 0.2567 | 0.3454 |
| | RH | 128.66 | 178.11 | 32.60 | NDCG@100 | 0.4342 | 0.3766 | 0.3187 |
| $\gamma = 100$ | LH | 60.75 | 86.54 | 15.85 | Recall@50 | 0.3453 | 0.2580 | 0.3472 |
| | RH | 125.90 | 176.25 | 32.26 | NDCG@100 | 0.4373 | 0.3785 | 0.3205 |
| $\gamma = 200$ | LH | 60.06 | 85.96 | 15.76 | Recall@50 | 0.3471 | 0.2592 | 0.3486 |
| | RH | 123.67 | 174.55 | 31.94 | NDCG@100 | 0.4402 | 0.3804 | 0.3220 |
| $\gamma = 500$ | LH | 59.46 | 85.35 | 15.66 | Recall@50 | 0.3489 | 0.2605 | 0.3490 |
| | RH | 121.41 | 172.64 | 31.62 | NDCG@100 | 0.4439 | 0.3826 | 0.3225 |
| $\gamma = 1000$ | LH | 59.19 | 85.00 | 15.64 | Recall@50 | 0.3502 | 0.2612 | 0.3475 |
| | RH | 120.17 | 171.44 | 31.50 | NDCG@100 | 0.4464 | 0.3840 | 0.3210 |
| $\gamma = 2000$ | LH | 59.09 | 84.72 | 15.68 | Recall@50 | 0.3510 | 0.2619 | 0.3434 |
| | RH | 119.34 | 170.45 | 31.52 | NDCG@100 | 0.4487 | 0.3854 | 0.3171 |
| $\gamma = 5000$ | LH | 59.19 | 84.48 | 15.83 | Recall@50 | 0.3506 | 0.2625 | 0.3340 |
| | RH | 118.91 | 169.47 | 31.77 | NDCG@100 | 0.4509 | 0.3871 | 0.3079 |

## Acknowledgments and Disclosure of Funding

This research was partially supported by NSF grant IIS 2142675. RG acknowledges the NeurIPS 2025 Financial Aid Award.

We thank all anonymous reviewers of the current and previous submissions of this manuscript for their valuable comments. In particular, the use of the relaxed MSE as a testing metric for LAEs was suggested by Reviewer nWHo, and the experimental comparison between our bound and practical ranking metrics was suggested by Reviewer TXNp.

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

# A  Proofs of Theorems, Lemmas, and Propositions

*Proof of Theorem 3.2*:

Given $W$ and $(x, y) \sim \mathcal{D}$, denote $v = y - Wx$, then $v \sim \mathcal{N}(\mu_W, \Sigma_W)$. Let $Q \in \mathbb{R}^{p \times p}$ such that $\Sigma_W = QQ^T$. Such $Q$ exists since we can take $Q = \Sigma_W^{1/2} = S^T \Lambda^{1/2} S$, but we do not assume it to be unique. Let $\epsilon \sim \mathcal{N}(0, I)$, then we can write $v = Q\epsilon + \mu_W$. Thus,

$$
\begin{aligned}
R^{\text{true}}(W) &= \mathbb{E}_{(x,y) \sim \mathcal{D}}\left[\|y - Wx\|_F^2\right] = \mathbb{E}_\epsilon\left[\|Q\epsilon + \mu_W\|_F^2\right] = \mathbb{E}_\epsilon\left[(Q\epsilon + \mu_W)^T(Q\epsilon + \mu_W)\right] \\
&= \mathbb{E}_\epsilon[\epsilon^T Q^T Q\epsilon + \mu_W^T Q\epsilon + \epsilon^T Q^T \mu_W + \mu_W^T \mu_W] = \text{tr}(Q^T Q) + \mu_W^T \mu_W \\
&= \text{tr}(QQ^T) + \mu_W^T \mu_W = \text{tr}(\Sigma_W) + \mu_W^T \mu_W
\end{aligned}
\tag{15}
$$

Also, we can express the random variable $\|v\|_F^2$ in quadratic form (Representation 3.1a.1, [36]):

$$
\begin{aligned}
\|v\|_F^2 = v^T v &= (Q\epsilon + \mu_W)^T(Q\epsilon + \mu_W) \\
&= (Q\epsilon + \mu_W)^T \Sigma_W^{-1/2} \Sigma_W \Sigma_W^{-1/2}(Q\epsilon + \mu_W) \\
&= (\Sigma_W^{-1/2} Q\epsilon + \Sigma_W^{-1/2}\mu_W)^T \Sigma_W (\Sigma_W^{-1/2} Q\epsilon + \Sigma_W^{-1/2}\mu_W) \\
&= (\Sigma_W^{-1/2} Q\epsilon + \Sigma_W^{-1/2}\mu_W)^T S^T \Lambda S (\Sigma_W^{-1/2} Q\epsilon + \Sigma_W^{-1/2}\mu_W) \\
&= (S\Sigma_W^{-1/2} Q\epsilon + S\Sigma_W^{-1/2}\mu_W)^T \Lambda (S\Sigma_W^{-1/2} Q\epsilon + S\Sigma_W^{-1/2}\mu_W)
\end{aligned}
\tag{16}
$$

Denote $\epsilon' = S\Sigma_W^{-1/2} Q\epsilon$, then $\epsilon' \sim \mathcal{N}(0, I)$, because $\mathbb{E}[\epsilon'] = S\Sigma_W^{-1/2} Q\mathbb{E}[\epsilon] = 0$ and

$$
\text{Cov}[\epsilon'] = \mathbb{E}[\epsilon'\epsilon'^T] = S\Sigma_W^{-1/2} Q\mathbb{E}[\epsilon\epsilon^T] Q^T \Sigma_W^{-1/2} S^T = I
$$

As $b = S\Sigma_W^{-1/2}\mu_W$, we can write

$$
\|v\|_F^2 = (\epsilon' + b)^T \Lambda (\epsilon' + b) = \sum_{i=1}^p \eta_i (\epsilon'_i + b_i)^2
$$

Hence, each $\epsilon'_i + b_i$ is independently from $\mathcal{N}(b_i, 1)$, and each $(\epsilon'_i + b_i)^2$ is independently from the non-central chi-squared distribution of noncentrality parameter $b_i^2$ and with degree 1 of freedom. Thus the MGF of $(\epsilon'_i + b_i)^2$ is

$$
M_{(\epsilon'_i + b_i)^2}(t) = \mathbb{E}_{(\epsilon'_i + b_i)^2}[e^{t(\epsilon'_i + b_i)^2}] = \frac{\exp\left(\frac{b_i^2 t}{1 - 2t}\right)}{(1 - 2t)^{1/2}}
\tag{17}
$$

Given i.i.d. samples $\{(x_j, y_j)\}_{j=1}^m$ from $\mathcal{D}$, let $v_j = y_j - Wx_j$. Then $v_1, v_2, ..., v_m$ are i.i.d. from $\mathcal{N}(\mu_W, \Sigma_W)$, and

$$
R^{\text{emp}}(W) = \frac{1}{m}\sum_{j=1}^m \|y_j - Wx_j\|_F^2 = \frac{1}{m}\sum_{j=1}^m \|v_j\|_F^2
$$

Hence the MGF of $R^{\text{emp}}(W)$ is

$$
\begin{aligned}
M_{R^{\text{emp}}(W)}(t) &= \mathbb{E}_{S \sim \mathcal{D}^m}\left[e^{tR^{\text{emp}}(W)}\right] = \mathbb{E}_{S \sim \mathcal{D}^m}\left[\exp\left(\frac{t}{m}\sum_{j=1}^m \|v_j\|_F^2\right)\right] \\
&= \left(\mathbb{E}_{S \sim \mathcal{D}^m}\left[\exp\left(\frac{t}{m}\|v\|_F^2\right)\right]\right)^m = \left(\mathbb{E}_{S \sim \mathcal{D}^m}\left[\exp\left(\frac{t}{m}\sum_{i=1}^p \eta_i(\epsilon'_i + b_i)^2\right)\right]\right)^m \\
&= \left(\prod_{i=1}^p \mathbb{E}_{(\epsilon'_i + b_i)^2}\left[\exp\left(\frac{t\eta_i}{m}(\epsilon'_i + b_i)^2\right)\right]\right)^m = \left(\prod_{i=1}^p \frac{\exp\left(\frac{tb_i^2\eta_i}{m - 2t\eta_i}\right)}{(1 - 2t\eta_i/m)^{1/2}}\right)^m \\
&= \frac{\exp\left(\sum_{i=1}^p \frac{tmb_i^2\eta_i}{m - 2t\eta_i}\right)}{\prod_{i=1}^p (1 - 2t\eta_i/m)^{m/2}}
\end{aligned}
\tag{18}
$$

By (15) and (18), we can expand $\Psi_{\pi,\mathcal{D}}(\lambda, m)$ as

$$\Psi_{\pi,\mathcal{D}}(\lambda, m) = \ln \mathbb{E}_{W \sim \pi} \mathbb{E}_{S \sim \mathcal{D}^m}[e^{\lambda(R^{\text{true}}(W) - R^{\text{emp}}(W))}]$$

$$= \ln \mathbb{E}_{W \sim \pi}\left[e^{\lambda R^{\text{true}}(W)} \mathbb{E}_{S \sim \mathcal{D}^m}[e^{-\lambda R^{\text{emp}}(W)}]\right]$$

$$= \ln \mathbb{E}_{W \sim \pi}\left[\exp\left(\lambda\left(\text{tr}(\Sigma_W) + \mu_W^T \mu_W\right)\right) \frac{\exp\left(\sum_{i=1}^p \frac{-\lambda m b_i^2 \eta_i}{m + 2\lambda \eta_i}\right)}{\prod_{i=1}^p (1 + 2\lambda \eta_i/m)^{m/2}}\right] \qquad (19)$$

Use the inequality that for any $x > 0$ and $k > 0$, $e^{\frac{xk}{x+k}} < (\frac{x}{k} + 1)^k$ [4], and the fact $\text{tr}(\Sigma_W) = \sum_{i=1}^p \eta_i$, we have

$$\ln \mathbb{E}_{W \sim \pi}\left[\exp\left(\lambda\left(\text{tr}(\Sigma_W) + \mu_W^T \mu_W\right)\right) \frac{\exp\left(\sum_{i=1}^p \frac{-\lambda m b_i^2 \eta_i}{m + 2\lambda \eta_i}\right)}{\prod_{i=1}^p (1 + 2\lambda \eta_i/m)^{m/2}}\right]$$

$$\leq \ln \mathbb{E}_{W \sim \pi}\left[\exp\left(\lambda\left(\text{tr}(\Sigma_W) + \mu_W^T \mu_W\right)\right) \frac{\exp\left(\sum_{i=1}^p \frac{-\lambda m b_i^2 \eta_i}{m + 2\lambda \eta_i}\right)}{\prod_{i=1}^p \exp\left(\frac{m\lambda\eta_i}{m + 2\lambda \eta_i}\right)}\right]$$

$$= \ln \mathbb{E}_{W \sim \pi} \exp\left(\lambda \mu_W^T \mu_W + \sum_{i=1}^p \lambda(\eta_i - \frac{m b_i^2 \eta_i}{m + 2\lambda \eta_i}) - \sum_{i=1}^p \frac{m\lambda\eta_i}{m + 2\lambda\eta_i}\right)$$

$$= \ln \mathbb{E}_{W \sim \pi} \exp\left(\lambda \mu_W^T \mu_W + \sum_{i=1}^p \frac{2\lambda^2 \eta_i^2 - \lambda m b_i^2 \eta_i}{m + 2\lambda\eta_i}\right)$$

$$\leq \ln \mathbb{E}_{W \sim \pi} \exp\left(\lambda(\mu_W^T \mu_W - \sum_{i=1}^p b_i^2 \eta_i) + \frac{2\lambda^2(\sum_{i=1}^p \eta_i^2)}{m}\right) = \ln \mathbb{E}_{W \sim \pi} \exp\left(\frac{2\lambda^2(\sum_{i=1}^p \eta_i^2)}{m}\right)$$

The last equality above is because

$$\sum_{i=1}^p b_i^2 \eta_i = b^T \Lambda b = \mu_W^T \Sigma_W^{-1/2} S^T \Lambda S \Sigma_W^{-1/2} \mu_W = \mu_W^T \mu_W$$

Since

$$\sum_{i=1}^p \eta_i^2 = \text{tr}(S^T \Lambda^2 S) = \text{tr}(\Sigma_W^2) = \text{tr}(\Sigma_W \Sigma_W^T) = \|\Sigma_W\|_F^2$$

we have

$$\ln \mathbb{E}_{W \sim \pi} \exp\left(\frac{2\lambda^2(\sum_{i=1}^p \eta_i^2)}{m}\right) = \ln \mathbb{E}_{W \sim \pi} \exp\left(\frac{2\lambda^2\|\Sigma_W\|_F^2}{m}\right)$$

$\square$

*Proof of Theorem 3.3*:

By (19), given any $\lambda > 0$, we let $\{f_m\}_{m \in \mathbb{N}}$ be a sequence of functions where

$$f_m(W) = \exp\left(\lambda\left(\text{tr}(\Sigma_W) + \mu_W^T \mu_W\right)\right) \frac{\exp\left(\sum_{i=1}^p \frac{-\lambda m b_i^2 \eta_i}{m + 2\lambda \eta_i}\right)}{\prod_{i=1}^p (1 + 2\lambda \eta_i/m)^{m/2}}$$

for $m > 0$, and

$$f_0(W) = \exp\left(\lambda\left(\text{tr}(\Sigma_W) + \mu_W^T \mu_W\right)\right)$$

Note that each $f_i$ is a non-negative function.

---

[4]Since $\frac{x}{x+1} < \ln(x + 1)$ for any $x > -1$, replacing $x$ with $\frac{x}{k}$, and taking exponential on both sides, we get $e^{\frac{xk}{x+k}} < (\frac{x}{k} + 1)^k$.

Since $W \sim \pi$ and $W \in \mathbb{R}^{p \times p}$, let $E = \mathbb{R}^{p \times p}$, then $f_m$ is a measurable function on $E$, $\pi$ is a Borel probability measure on $E$, and we can express $\mathbb{E}_\pi [f_m]$ as a Lebesgue integral (Section 10.1, [15]):

$$\mathbb{E}_\pi [f_m] = \int_E f_m \, d\pi$$

Now we prove the following three conditions:

(a) $f_m(W) \leq f_0(W)$ for any $m \geq 0$ and $W \in E$.

For any given $W$, $\eta_i$ and $b_i$ are fixed for all $i$, where $\eta_i > 0$ and $b_i^2 \geq 0$. Thus for any $m \geq 0$, the numerator of $f_m$ satisfies $\exp \left( \sum_{i=1}^p \frac{-\lambda m b_i^2 \eta_i}{m + 2\lambda \eta_i} \right) \leq 1$, and the denominator of $f_m$ satisfies $\prod_{i=1}^p (1 + 2\lambda \eta_i / m)^{m/2} \geq 1$. Note that the numerator is monotonically decreasing with $m$ and the denominator is monotonically increasing with $m$.

(b) $f_m \to 1$ pointwisely as $m \to \infty$.

For any $W$,

$$\lim_{m \to \infty} f_m(W) = \exp \left( \lambda \left( \mathrm{tr}(\Sigma_w) + \mu_w^T \mu_w \right) \right) \lim_{m \to \infty} \frac{\exp \left( \sum_{i=1}^p \frac{-\lambda m b_i^2 \eta_i}{m + 2\lambda \eta_i} \right)}{\prod_{i=1}^p (1 + 2\lambda \eta_i / m)^{m/2}}$$

$$= \exp \left( \lambda \left( \mathrm{tr}(\Sigma_w) + \mu_w^T \mu_w \right) \right) \frac{\exp \left( \sum_{i=1}^p \lim_{m \to \infty} \frac{-\lambda m b_i^2 \eta_i}{m + 2\lambda \eta_i} \right)}{\prod_{i=1}^p \lim_{m \to \infty} (1 + 2\lambda \eta_i / m)^{m/2}}$$

$$= \exp \left( \lambda \left( \mathrm{tr}(\Sigma_w) + \mu_w^T \mu_w \right) \right) \frac{\exp \left( -\lambda \sum_{i=1}^p b_i^2 \eta_i \right)}{\prod_{i=1}^p \exp (\lambda \eta_i)} = 1$$

The last inequality uses the facts that $\sum_{i=1}^p b_i^2 \eta_i = \mu_w^T \mu_w$ and $\sum_{i=1}^p \eta_i = \mathrm{tr}(\Sigma_w)$.

(c) $\int_E f_0 \, d\pi = \mathbb{E}_\pi[f_0] < \infty$.

$$\mathbb{E}_\pi[f_0] = \mathbb{E}_\pi \exp \left( \lambda \left( \mathrm{tr}(\Sigma_w) + \mu_w^T \mu_w \right) \right)$$

$$= \mathbb{E}_\pi \exp \left( \lambda \left[ \mathrm{tr}((W^* - W)\Sigma_x(W^* - W)^T + \Sigma_e) + \|(W^* - W)\mu_x\|_F^2 \right] \right)$$

$$= \mathbb{E}_\pi \exp \left( \lambda \left[ \sum_{i=1}^p (W^* - W)_{i*} \Sigma_x (W^* - W)_{i*}^T + \mathrm{tr}(\Sigma_e) + \sum_{i=1}^p (W^* - W)_{i*} \mu_x \mu_x^T (W^* - W)_{i*}^T \right] \right)$$

$$= \mathbb{E}_\pi \exp \left( \lambda \left[ \sum_{i=1}^p (W^* - W)_{i*} \left[ \Sigma_x + \mu_x \mu_x^T \right] (W^* - W)_{i*}^T + \mathrm{tr}(\Sigma_e) \right] \right)$$

$$= \mathbb{E}_\pi \exp \left( \lambda \left[ \left\| \left( \Sigma_x + \mu_x \mu_x^T \right)^{1/2} (W^* - W) \right\|_F^2 + \mathrm{tr}(\Sigma_e) \right] \right)$$

$$= \exp \left( \lambda \mathrm{tr}(\Sigma_e) \right) \mathbb{E}_\pi \exp \left( \lambda \left[ \left\| \left( \Sigma_x + \mu_x \mu_x^T \right)^{1/2} (W^* - W) \right\|_F^2 \right] \right) < \infty$$

The last inequality holds because $\mathbb{E}_\pi \exp \left( \lambda \left[ \left\| \left( \Sigma_x + \mu_x \mu_x^T \right)^{1/2} (W^* - W) \right\|_F^2 \right] \right) < \infty$ is our assumption and $\exp \left( \lambda \mathrm{tr}(\Sigma_e) \right)$ is a constant.

Since the conditions (a), (b) and (c) hold, by the *dominated convergence theorem* (Theorem 11.32, [44]), we have

$$\lim_{m \to \infty} \mathbb{E}_\pi [f_m] = \lim_{m \to \infty} \int_E f_m \, d\pi = \int_E \lim_{m \to \infty} f_m \, d\pi = \int_E 1 \, d\pi = \mathbb{E}_\pi[1] = 1$$

Since $\ln$ is continuous on $(0, \infty)$, we can interchange $\lim$ and $\ln$. Therefore,

$$\lim_{m \to \infty} \Psi_{\pi, \mathcal{D}}(\lambda, m) = \lim_{m \to \infty} \ln \mathbb{E}_\pi[f_m] = \ln \lim_{m \to \infty} \mathbb{E}_\pi[f_m] = \ln 1 = 0$$

$\square$

*Proof of Lemma 4.2*:

$$R^{\text{true}}(W) = \mathbb{E}\left[\|y - Wx\|_F^2\right] = \sum_{i=1}^n \mathbb{E}[\|y_i - W_{i*}x\|_F^2] = \sum_{i=1}^n W_{i*}\mathbb{E}[xx^T]W_{i*}^T - 2W_{i*}\mathbb{E}[y_ix] + \mathbb{E}[y_i^2]$$

$$= \sum_{i=1}^n W_{i*}\Sigma_{xx}W_{i*}^T - 2W_{i*}(\Sigma_{xy})_{*i} + (\Sigma_{yy})_{ii}$$

$$= \sum_{i=1}^n (W_{i*}\Sigma_{xx}^{1/2})(W_{i*}\Sigma_{xx}^{1/2})^T - 2(W_{i*}\Sigma_{xx}^{1/2})\Sigma_{xx}^{-1/2}(\Sigma_{xy})_{*i} + (\Sigma_{yy})_{ii}$$

$$= \sum_{i=1}^n (W_{i*}\Sigma_{xx}^{1/2} - (\Sigma_{xy})_{*i}^T\Sigma_{xx}^{-1/2})(W_{i*}\Sigma_{xx}^{1/2} - (\Sigma_{xy})_{*i}^T\Sigma_{xx}^{-1/2})^T - (\Sigma_{xy})_{*i}^T\Sigma_{xx}^{-1}(\Sigma_{xy})_{*i} + (\Sigma_{yy})_{ii}$$

$$= \sum_{i=1}^n \|W_{i*}\Sigma_{xx}^{1/2} - (\Sigma_{xy})_{*i}^T\Sigma_{xx}^{-1/2}\|_F^2 - \|\Sigma_{xx}^{-1/2}(\Sigma_{xy})_{*i}\|_F^2 + (\Sigma_{yy})_{ii}$$

$$= \|W\Sigma_{xx}^{1/2} - \Sigma_{xy}^T\Sigma_{xx}^{-1/2}\|_F^2 - \|\Sigma_{xy}^T\Sigma_{xx}^{-1/2}\|_F^2 + \text{tr}(\Sigma_{yy})$$

Since we assume $\Sigma_{xx}$ is positive definite, $\Sigma_{xx}^{-1/2}$ exists.

$\square$

*Proof of Proposition 4.3*:

By Lemma 4.2,

$$\mathbb{E}_\pi\left[e^{\lambda R^{\text{true}}(W)}\right] = \mathbb{E}_\pi\left[e^{\lambda\left(\|W\Sigma_{xx}^{1/2} - \Sigma_{xy}^T\Sigma_{xx}^{-1/2}\|_F^2 - \|\Sigma_{xy}^T\Sigma_{xx}^{-1/2}\|_F^2 + \text{tr}(\Sigma_{yy})\right)}\right]$$

$$= \mathbb{E}_\pi\left[e^{\lambda\|W\Sigma_{xx}^{1/2} - \Sigma_{xy}^T\Sigma_{xx}^{-1/2}\|_F^2}\right]e^{\lambda\left(\text{tr}(\Sigma_{yy}) - \|\Sigma_{xy}^T\Sigma_{xx}^{-1/2}\|_F^2\right)} = C\,\mathbb{E}_\pi\left[e^{\lambda\|W\Sigma_{xx}^{1/2} + B\|_F^2}\right] \qquad (20)$$

$\square$

*Proof of Lemma 4.4*:

Since $x = \boldsymbol{\delta} \odot h$ and $y = (\mathbf{1} - \boldsymbol{\delta}) \odot h$, we have

$$\Sigma_{xx} = \mathbb{E}\left[xx^T\right] = \mathbb{E}\left[(\boldsymbol{\delta} \odot h)(\boldsymbol{\delta} \odot h)^T\right]$$
$$\Sigma_{xy} = \mathbb{E}\left[xy^T\right] = \mathbb{E}\left[(\boldsymbol{\delta} \odot h)((\mathbf{1} - \boldsymbol{\delta}) \odot h)^T\right]$$
$$\Sigma_{yy} = \mathbb{E}\left[yy^T\right] = \mathbb{E}\left[((\mathbf{1} - \boldsymbol{\delta}) \odot h)((\mathbf{1} - \boldsymbol{\delta}) \odot h)^T\right]$$

We first prove $\Sigma_{xx}$. For $i, j \in \{1, 2, ..., n\}$ with $i \neq j$, $(\Sigma_{xx})_{ij} = \mathbb{E}[\boldsymbol{\delta}_i\boldsymbol{\delta}_jh_ih_j]$. Note that $\boldsymbol{\delta}_i\boldsymbol{\delta}_jh_ih_j$ is a Bernoulli random variable (as its value can either be 0 or 1), $\boldsymbol{\delta}_i$ depends on $h_i$, and $\boldsymbol{\delta}_j$ depends on $h_j$. We have

$$\mathbb{E}[\boldsymbol{\delta}_i\boldsymbol{\delta}_jh_ih_j] = P\left(\boldsymbol{\delta}_i\boldsymbol{\delta}_jh_ih_j = 1\right) = P\left(\boldsymbol{\delta}_i = 1, \boldsymbol{\delta}_j = 1, h_i = 1, h_j = 1\right)$$
$$= P\left(\boldsymbol{\delta}_i = 1|\boldsymbol{\delta}_j = 1, h_i = 1, h_j = 1\right)P\left(\boldsymbol{\delta}_j = 1|h_i = 1, h_j = 1\right)P\left(h_i = 1, h_j = 1\right)$$
$$= P\left(\boldsymbol{\delta}_i = 1|h_i = 1\right)P\left(\boldsymbol{\delta}_j = 1|h_j = 1\right)P\left(h_i = 1, h_j = 1\right)$$
$$= p^2\mathbb{E}\left[h_ih_j\right] = p^2(\Sigma_{hh})_{ij} \qquad (21)$$

For any $i$, $(\Sigma_{xx})_{ii} = \mathbb{E}[(\boldsymbol{\delta}_ih_i)^2]$. Using the property that a Bernoulli random variable $X$ has $\mathbb{E}[X^2] = \mathbb{E}[X]$,

$$\mathbb{E}[(\boldsymbol{\delta}_ih_i)^2] = P\left(\boldsymbol{\delta}_ih_i = 1\right) = P\left(\boldsymbol{\delta}_i = 1, h_i = 1\right) = P\left(\boldsymbol{\delta}_i = 1|h_i = 1\right)P\left(h_i = 1\right) = p\mathbb{E}[h_i]$$
$$= p\mathbb{E}[h_i^2] = p(\Sigma_{hh})_{ii} \qquad (22)$$

Combining (21) and (22), we get

$$\Sigma_{xx} = p^2\Sigma_{hh} + p(1 - p)(I \odot \Sigma_{hh}) \qquad (23)$$

Since $(\Sigma_{yy})_{ij} = \mathbb{E}[(1 - \boldsymbol{\delta}_i)(1 - \boldsymbol{\delta}_j)h_i h_j]$ and $(\Sigma_{yy})_{ii} = \mathbb{E}[((1 - \boldsymbol{\delta}_i)h_i)^2]$, replacing $p$ with $1 - p$ in (23), we get $\Sigma_{yy} = (1 - p)^2 \Sigma_{hh} + p(1 - p)(I \odot \Sigma_{hh})$.

Since $(\Sigma_{xy})_{ij} = \mathbb{E}[\boldsymbol{\delta}_i(1 - \boldsymbol{\delta}_j)h_i h_j] = p(1 - p)\Sigma_{hh}$ and $(\Sigma_{xy})_{ii} = \mathbb{E}[\boldsymbol{\delta}_i(1 - \boldsymbol{\delta}_i)h_i^2] = 0$ (Note that $\boldsymbol{\delta}_i(1 - \boldsymbol{\delta}_i)h_i^2 = 0$ regardless of whether $\boldsymbol{\delta}_i$ is 0 or 1.), we have $\Sigma_{xy} = p(1 - p)(\Sigma_{hh} - I \odot \Sigma_{hh})$.

Note that in (23), if $\Sigma_{hh}$ is positive definite, then $I \odot \Sigma_{hh}$ is also positive definite (Theorem 7.5.3 (b), [22]), which implies that $\Sigma_{xx}$ is positive definite.

$\square$

*Proof of Theorem 5.2*:

*(a)* Since $\mathbb{E}_\rho[W] = \mathcal{U}$ and $\mathbb{E}_\rho[W^T W] = \mathcal{U}^T \mathcal{U} + \mathrm{diag}\left(\sum_{k=1}^n \mathcal{S}_{k1}, \sum_{k=1}^n \mathcal{S}_{k2}, ..., \sum_{k=1}^n \mathcal{S}_{kn}\right)$,

$$\mathbb{E}_\rho[R^{\mathrm{emp}}(W)] = \frac{1}{m}\mathbb{E}_\rho[\|Y - WX\|_F^2] = \frac{1}{m}\sum_{l=1}^m \mathbb{E}_\rho[\|Y_{*l} - WX_{*l}\|_F^2]$$

$$= \frac{1}{m}\sum_{l=1}^m \mathbb{E}_\rho[(Y_{*l} - WX_{*l})^T(Y_{*l} - WX_{*l})] = \frac{1}{m}\sum_{l=1}^m Y_{*l}^T Y_{*l} - 2Y_{*l}^T \mathbb{E}_\rho[W] X_{*l} + X_{*l}^T \mathbb{E}_\rho[W^T W] X_{*l}$$

$$= \frac{1}{m}\sum_{l=1}^m Y_{*l}^T Y_{*l} - 2Y_{*l}^T \mathcal{U} X_{*l} + X_{*l}^T \mathcal{U}^T \mathcal{U} X_{*l} + X_{*l}^T \mathrm{diag}\left(\sum_{k=1}^n \mathcal{S}_{k1}, \sum_{k=1}^n \mathcal{S}_{k2}, ..., \sum_{k=1}^n \mathcal{S}_{kn}\right) X_{*l}$$

$$\tag{24}$$

And $D(\rho \| \pi)$ can be expressed as

$$D(\rho \| \pi) = \frac{1}{2}\left[n^2(2\ln\sigma - 1) - \sum_{k=1}^n \sum_{l=1}^n (\ln \mathcal{S}_{kl} - \frac{\mathcal{S}_{kl}}{\sigma^2}) + \frac{\|\mathcal{U} - \mathcal{U}_0\|_F^2}{\sigma^2}\right] \tag{25}$$

Denote $f(\mathcal{U}, \mathcal{S}|\mathcal{U}_0, \sigma, \lambda) = \mathbb{E}_\rho[R^{\mathrm{emp}}(W)] + \frac{1}{\lambda}D(\rho \| \pi)$, our optimization problem becomes

$$\min_{\mathcal{U}, \mathcal{S}} f(\mathcal{U}, \mathcal{S}|\mathcal{U}_0, \sigma, \lambda) \tag{26}$$

The optimal $\mathcal{U}$ and $\mathcal{S}$ are obtained by solving $\frac{\partial}{\partial \mathcal{U}} f(\mathcal{U}, \mathcal{S}|\mathcal{U}_0, \sigma, \lambda) = 0$ and $\frac{\partial}{\partial \mathcal{S}} f(\mathcal{U}, \mathcal{S}|\mathcal{U}_0, \sigma, \lambda) = 0$.

First we show the partial derivatives of the $\frac{1}{\lambda}D(\rho \| \pi)$ term:

$$\frac{\partial}{\partial \mathcal{U}_{ij}} \frac{1}{\lambda}D(\rho \| \pi) = \frac{(\mathcal{U}_{ij} - (\mathcal{U}_0)_{ij})}{\lambda\sigma^2}, \quad \frac{\partial}{\partial \mathcal{S}_{ij}} \frac{1}{\lambda}D(\rho \| \pi) = -\frac{1}{2\lambda}(\frac{1}{\mathcal{S}_{ij}} - \frac{1}{\sigma^2})$$

Then we show the partial derivatives of the $\mathbb{E}_\rho[R^{\mathrm{emp}}(W)]$ term. By (24), for any $i, j$,

$$\frac{\partial}{\partial \mathcal{S}_{ij}}\mathbb{E}_\rho[R^{\mathrm{emp}}(W)] = \frac{\partial}{\partial \mathcal{S}_{ij}} \frac{1}{m}\sum_{l=1}^m X_{*l}^T \mathrm{diag}\left(\sum_{k=1}^n \mathcal{S}_{k1}, \sum_{k=1}^n \mathcal{S}_{k2}, ..., \sum_{k=1}^n \mathcal{S}_{kn}\right) X_{*l}$$

$$= \frac{\partial}{\partial \mathcal{S}_{ij}} \frac{1}{m}\sum_{l=1}^m X_{jl}\mathcal{S}_{ij}X_{jl} = \frac{1}{m}\sum_{l=1}^m X_{jl}^2 = \frac{1}{m}X_{j*}X_{j*}^T$$

$$\frac{\partial}{\partial \mathcal{U}_{ij}}\mathbb{E}_\rho[R^{\mathrm{emp}}(W)] = \frac{\partial}{\partial \mathcal{U}_{ij}} \frac{1}{m}\sum_{l=1}^m -2Y_{*l}^T \mathcal{U} X_{*l} + X_{*l}^T \mathcal{U}^T \mathcal{U} X_{*l} = \frac{1}{m}\sum_{l=1}^m \left(-2Y_{il}X_{jl} + \frac{\partial}{\partial \mathcal{U}_{ij}}\sum_{k=1}^n (\mathcal{U}_{k*}X_{*l})^2\right)$$

$$= \frac{1}{m}\sum_{l=1}^m \left(-2Y_{il}X_{jl} + \frac{\partial}{\partial \mathcal{U}_{ij}}(\mathcal{U}_{i*}X_{*l})^2\right) = \frac{1}{m}\sum_{l=1}^m (-2Y_{il}X_{jl} + 2(\mathcal{U}_{i*}X_{*l})X_{jl})$$

$$= \frac{2}{m}\left(-Y_{i*}X_{j*}^T + \mathcal{U}_{i*}XX_{j*}^T\right)$$

Wrap up the above results, we get

$$\frac{\partial}{\partial \mathcal{S}_{ij}} f(\mathcal{U}, \mathcal{S}|\mathcal{U}_0, \sigma, \lambda) = \frac{1}{m}X_{j*}X_{j*}^T - \frac{1}{2\lambda}(\frac{1}{\mathcal{S}_{ij}} - \frac{1}{\sigma^2}) \tag{27}$$

$$\frac{\partial}{\partial \mathcal{U}_{ij}} f(\mathcal{U}, \mathcal{S}|\mathcal{U}_0, \sigma, \lambda) = \frac{2}{m}\left(-Y_{i*}X_{j*}^T + \mathcal{U}_{i*}XX_{j*}^T\right) + \frac{(\mathcal{U}_{ij} - (\mathcal{U}_0)_{ij})}{\lambda\sigma^2} \tag{28}$$

Therefore, by (27), the solution of $\frac{\partial}{\partial \mathcal{S}} f(\mathcal{U}, \mathcal{S}|\mathcal{U}_0, \sigma, \lambda) = 0$ is that

$$\mathcal{S}_{ij} = \frac{1}{\frac{2\lambda}{m}X_{j*}X_{j*}^T + \frac{1}{\sigma^2}} \quad \text{for any } i, j \in \{1, 2, ..., n\} \tag{29}$$

By (28) we have

$$\frac{\partial}{\partial \mathcal{U}} f(\mathcal{U}, \mathcal{S}|\mathcal{U}_0, \sigma, \lambda) = \left[\frac{2}{m}(-YX^T + \mathcal{U}XX^T) + \frac{1}{\lambda\sigma^2}(\mathcal{U} - \mathcal{U}^0)\right]^T \tag{30}$$

Thus the solution of $\frac{\partial}{\partial \mathcal{U}} f(\mathcal{U}, \mathcal{S}|\mathcal{U}_0, \sigma, \lambda) = 0$ is

$$\mathcal{U} = \left(\frac{1}{m}YX^T + \frac{1}{2\lambda\sigma^2}\mathcal{U}_0\right)\left(\frac{1}{m}XX^T + \frac{1}{2\lambda\sigma^2}I\right)^{-1} \tag{31}$$

Now we show that $f(\mathcal{U}, \mathcal{S}|\mathcal{U}_0, \sigma, \lambda)$ is a convex function, such that the solutions of $\mathcal{S}$ in (29) and $\mathcal{U}$ in (31) are the global minimizer of (26). By (27) and (28) we have

$$\frac{\partial^2 f}{\partial \mathcal{S}_{ij}\partial \mathcal{S}_{kl}} = \begin{cases} \frac{1}{2\lambda(\mathcal{S}_{ij})^2} & \text{if } i = k, j = l \\ 0 & \text{otherwise} \end{cases}, \quad \frac{\partial^2 f}{\partial \mathcal{U}_{ij}\partial \mathcal{U}_{kl}} = \begin{cases} \frac{2}{m}X_{j*}X_{l*}^T + \frac{1}{\lambda\sigma^2} & \text{if } i = k, j = l \\ \frac{2}{m}X_{j*}X_{l*}^T & \text{if } i = k, j \neq l \\ 0 & \text{otherwise} \end{cases}$$

Denote $\nu \in \mathbb{R}^{2n^2}$ where for $i = 1, 2, ..., n$ and $j = 1, 2, ..., n$, $\nu_{(i-1)n+j} = \mathcal{U}_{ij}$ and $\nu_{n^2+(i-1)n+j} = \mathcal{S}_{ij}$. Let $H_f \in \mathbb{R}^{2n^2 \times 2n^2}$ be the Hessian matrix where $(H_f)_{ij} = \frac{\partial^2 f}{\partial \nu_i \partial \nu_j}$. Then we can write $H_f = \begin{bmatrix} A & 0 \\ 0 & B \end{bmatrix}$ where $A = \frac{2}{m}(XX^T) \otimes I_n + \frac{1}{\lambda\sigma^2}I_{n^2}$ and $B$ is a $n^2 \times n^2$ diagonal matrix with $B_{(i-1)n+j,(i-1)n+j} = \frac{1}{2\lambda(\mathcal{S}_{ij})^2}$. Here $\otimes$ means Kronecker product.

The Kronecker product has a property that, let $\{\lambda_i | i = 1, ..., m\}$ be the eigenvalues of $P \in \mathbb{R}^{m \times m}$ and $\{\mu_j | j = 1, ..., n\}$ be the eigenvalues of $Q \in \mathbb{R}^{n \times n}$, then $\{\lambda_i \mu_j | i = 1, ..., m, j = 1, ..., n\}$ are the eigenvalues of $P \otimes Q$ (Theorem 4.2.12, [21]). Since $XX^T$ is positive semi-definite and $I_n$ is positive definite, $(XX^T) \otimes I_n$ is positive semi-definite. Thus $A$ is positive definite. Since all elements of $\mathcal{S}$ are positive, $B$ is positive definite. Therefore, $H_f$ is a positive definite matrix for any $\mathcal{U}$ and $\mathcal{S}$, which means $f(\mathcal{U}, \mathcal{S}|\mathcal{U}_0, \sigma, \lambda)$ is a convex function. Thus, the solutions of $\mathcal{S}$ in (29) and $\mathcal{U}$ in (31) give the global minimum.

*(b)* Applying the constraint $\text{diag}(W) = 0$ to $\rho$ and $\pi$ implies taking $\text{diag}(\mathcal{U}) = 0$, $\text{diag}(\mathcal{S}) = 0$, $\text{diag}(\mathcal{U}_0) = 0$, and $\text{diag}(\sigma^2 J) = 0$. In this case, (26) becomes a constrained optimization problem.

$$\min_{\mathcal{U}, \mathcal{S}} f(\mathcal{U}, \mathcal{S}|\mathcal{U}_0, \sigma, \lambda) \quad \text{s.t. } \text{diag}(\mathcal{U}) = 0, \ \text{diag}(\mathcal{S}) = 0 \tag{32}$$

Then we remove the constraint $\text{diag}(\mathcal{S}) = 0$ by defining $f$ as a function of only the off-diagonal elements of $\mathcal{S}$. Let $\mathcal{S}^- = \{\mathcal{S}_{ij} : i, j \in \{1, ..., n\}, i \neq j\}$, then (32) is equivalent to

$$\min_{\mathcal{U}, \mathcal{S}^-} f(\mathcal{U}, \mathcal{S}^-|\mathcal{U}_0, \sigma, \lambda) \quad \text{s.t. } \text{diag}(\mathcal{U}) = 0 \tag{33}$$

To solve (33), we construct the Lagrangian function

$$L(\mathcal{U}, \mathcal{S}^-, x|\mathcal{U}_0, \sigma, \lambda) = f(\mathcal{U}, \mathcal{S}^-|\mathcal{U}_0, \sigma, \lambda) + x^T \text{diag}(\mathcal{U})$$

where $x \in \mathbb{R}^n$, and solve

$$\frac{\partial L}{\partial x} = [\text{diag}(\mathcal{U})]^T = 0 \tag{34}$$

$$\frac{\partial L}{\partial \mathcal{U}} = \frac{\partial}{\partial \mathcal{U}} f(\mathcal{U}, \mathcal{S}^-|\mathcal{U}_0, \sigma, \lambda) + \text{diag}(x) = 0 \tag{35}$$

$$\frac{\partial L}{\partial \mathcal{S}_{ij}} = \frac{\partial}{\partial \mathcal{S}_{ij}} f(\mathcal{U}, \mathcal{S}^-|\mathcal{U}_0, \sigma, \lambda) = 0 \quad \text{for } i, j \in \{1, 2, ..., n\}, i \neq j \tag{36}$$

Since (36) is the $i \neq j$ case of (27), the optimal $\mathcal{S}^-$ is obtained by (29) with $i \neq j$.

The optimal $\mathcal{U}$ is obtained by solving (35) and (34). By (35),

$$\frac{2}{m}(-YX^T + \mathcal{U}XX^T) + \frac{1}{\lambda\sigma^2}(\mathcal{U} - \mathcal{U}^0) + \mathrm{diag}(x) = 0$$

$$\Longleftrightarrow \mathcal{U} = \left(\frac{1}{m}YX^T + \frac{1}{2\lambda\sigma^2}\mathcal{U}_0 - \frac{1}{2}\mathrm{diag}(x)\right)\left(\frac{1}{m}XX^T + \frac{1}{2\lambda\sigma^2}I\right)^{-1} \qquad (37)$$

Then we solve $x$ to satisfy (34),

$$\mathrm{diag}(\mathcal{U}) = \mathrm{diag}\left[\left(\frac{1}{m}YX^T + \frac{1}{2\lambda\sigma^2}\mathcal{U}_0\right)\left(\frac{1}{m}XX^T + \frac{1}{2\lambda\sigma^2}I\right)^{-1}\right] - \mathrm{diag}\left[\frac{1}{2}\mathrm{diag}(x)\left(\frac{1}{m}XX^T + \frac{1}{2\lambda\sigma^2}I\right)^{-1}\right]$$

$$= \mathrm{diag}\left[\left(\frac{1}{m}YX^T + \frac{1}{2\lambda\sigma^2}\mathcal{U}_0\right)\left(\frac{1}{m}XX^T + \frac{1}{2\lambda\sigma^2}I\right)^{-1}\right] - \frac{1}{2}x \odot \mathrm{diag}\left[\left(\frac{1}{m}XX^T + \frac{1}{2\lambda\sigma^2}I\right)^{-1}\right] = 0$$

we get

$$x = 2 \cdot \mathrm{diag}\left[\left(\frac{1}{m}YX^T + \frac{1}{2\lambda\sigma^2}\mathcal{U}_0\right)\left(\frac{1}{m}XX^T + \frac{1}{2\lambda\sigma^2}I\right)^{-1}\right] \oslash \mathrm{diag}\left[\left(\frac{1}{m}XX^T + \frac{1}{2\lambda\sigma^2}I\right)^{-1}\right]$$

Now we show that the solution of (34), (35) and (36) gives the global minimum of the problem (33). Let $H_L$ be the Hessian matrix of the Lagrangian $L$. It is easy to verify that, by removing the rows and columns of $H_f$ corresponding to $\mathcal{S}_{11}, \mathcal{S}_{22}, ...\mathcal{S}_{nn}$ to obtain $H'_f \in \mathbb{R}^{(2n^2-n)\times(2n^2-n)}$, we have $H_L = H'_f$. This shows that $H_L$ is positive definite for any $\mathcal{U}$ and $\mathcal{S}^-$. Hence, by the second-order sufficiency conditions (Section 11.5, [33]), any solution $(\mathcal{U}, \mathcal{S}^-, x)$ satisfying (34), (35) and (36) is a strict local minimum. Since the solution is unique, it is also a strict global minimum.

$\square$

*Proof of Proposition 5.3*:

Denote $v_i = (W_{i*}\Sigma_{xx}^{1/2} + B_{i*})^T$, and write $v_i = A^{1/2}\epsilon + \mu^i$ where $\epsilon \sim \mathcal{N}(0, I)$. Since $A = S^T\Lambda S$, using the quadratic form shown in (16), we have

$$\|v_i\|_F^2 = (A^{1/2}\epsilon + \mu^i)^T(A^{1/2}\epsilon + \mu^i) = (A^{1/2}\epsilon + \mu^i)^T A^{-1/2}S^T\Lambda S A^{-1/2}(A^{1/2}\epsilon + \mu^i)$$

$$= (S\epsilon + SA^{-1/2}\mu^i)^T\Lambda(S\epsilon + SA^{-1/2}\mu^i) = (S\epsilon + \bar{b}^i)^T\Lambda(S\epsilon + \bar{b}^i) = \sum_{j=1}^n \eta_j(S_{j*}\epsilon + \bar{b}_j^i)^2$$

It is easy to show that each $S_{j*}\epsilon$ are i.i.d. from $\mathcal{N}(0, 1)$ for all $j$, thus each $S_{j*}\epsilon + \bar{b}_j^i$ is independently from $\mathcal{N}(\bar{b}_j^i, 1)$. Since each $v_i$ is independent, by (20) we have

$$\mathbb{E}_\pi\left[e^{\lambda R^{\mathrm{true}}(W)}\right] = C\,\mathbb{E}_\pi\left[e^{\lambda\sum_{i=1}^n \|W_{i*}\Sigma_{xx}^{1/2}+B_{i*}\|_F^2}\right] = C\prod_{i=1}^n \mathbb{E}_\pi\left[e^{\lambda\|v_i\|_F^2}\right] = C\prod_{i=1}^n\prod_{j=1}^n \mathbb{E}_\pi\left[e^{\lambda\eta_j(S_{j*}\epsilon+\bar{b}_j^i)^2}\right]$$

$$= C\prod_{i=1}^n\prod_{j=1}^n \frac{\exp\left(\frac{\lambda(\bar{b}_j^i)^2\eta_j}{1-2\lambda\eta_j}\right)}{(1-2\lambda\eta_j)^{1/2}}$$

The last equality above follows from (17).

$\square$

*Proof of Theorem 5.4*:

Let $P, Q \in \mathbb{R}^{n\times n}$ be two symmetric matrices, we write $P \succeq Q$ if $P - Q$ is positive semi-definite, and write $P \succ Q$ if $P - Q$ is positive definite.

By Corollary 7.7.4 (c) of [22], if $P \succeq Q$, then $\eta_j(P) \geq \eta_j(Q)$ for any $j$, where $\eta_j(P)$ and $\eta_j(Q)$ denote the $j$th largest eigenvalues of $P$ and $Q$, respectively. Since $A - A^{(i)} = \sigma^2(\Sigma_{xx}^{1/2})_{*i}(\Sigma_{xx}^{1/2})_{*i}^T \succeq 0$ for any $i$, we have $\eta_j \geq \eta_j^{(i)}$ for any $i, j$.

Since $b^{(i)} = S^{(i)}(A^{(i)})^{-1/2}\mu^i$, we have

$$
\begin{aligned}
(b_j^{(i)})^2\eta_j^{(i)} &= \eta_j^{(i)}(\mu^i)^T(A^{(i)})^{-1/2}(S_{j*}^{(i)})^T S_{j*}^{(i)}(A^{(i)})^{-1/2}\mu^i \\
&= \eta_j^{(i)}(\mu^i)^T(S^{(i)})^T(\Lambda^{(i)})^{-1/2}[S^{(i)}(S_{j*}^{(i)})^T][S_{j*}^{(i)}(S^{(i)})^T](\Lambda^{(i)})^{-1/2}(S^{(i)})\mu^i \\
&= (\mu^i)^T(S_{j*}^{(i)})^T(S_{j*}^{(i)})\mu^i
\end{aligned}
$$

Therefore, (13) can be expressed as

$$
\frac{1}{C}\,\mathbb{E}_{\pi'}\left[e^{\lambda R^{\text{true}}(W)}\right] = \prod_{i=1}^{n}\prod_{j=1}^{n}\frac{\exp\left(\frac{\lambda(b_j^{(i)})^2\eta_j^{(i)}}{1-2\lambda\eta_j^{(i)}}\right)}{\left(1-2\lambda\eta_j^{(i)}\right)^{1/2}} = \prod_{i=1}^{n}\prod_{j=1}^{n}\frac{\exp\left(\frac{\lambda(\mu^i)^T(S_{j*}^{(i)})^T(S_{j*}^{(i)})\mu^i}{1-2\lambda\eta_j^{(i)}}\right)}{\left(1-2\lambda\eta_j^{(i)}\right)^{1/2}}
$$

$$
= \prod_{i=1}^{n}\frac{\exp\left(\lambda(\mu^i)^T\left(\sum_{j=1}^{n}\frac{(S_{j*}^{(i)})^T(S_{j*}^{(i)})}{1-2\lambda\eta_j^{(i)}}\right)\mu^i\right)}{\prod_{j=1}^{n}\left(1-2\lambda\eta_j^{(i)}\right)^{1/2}} = \prod_{i=1}^{n}\frac{\exp\left(\lambda(\mu^i)^T(S^{(i)})^T\bar{\Lambda}^{(i)}S^{(i)}\mu^i\right)}{\prod_{j=1}^{n}\left(1-2\lambda\eta_j^{(i)}\right)^{1/2}}
$$

where $\bar{\Lambda}^{(i)} = \text{diag}\left(\frac{1}{1-2\lambda\eta_1^{(i)}}, \frac{1}{1-2\lambda\eta_2^{(i)}}, ...., \frac{1}{1-2\lambda\eta_n^{(i)}}\right)$.

Similarly, (12) can be expressed as

$$
\frac{1}{C}\,\mathbb{E}_{\pi}\left[e^{\lambda R^{\text{true}}(W)}\right] = \prod_{i=1}^{n}\frac{\exp\left(\lambda(\mu^i)^T S^T\bar{\Lambda}S\mu^i\right)}{\prod_{j=1}^{n}\left(1-2\lambda\eta_j\right)^{1/2}}
$$

where $\bar{\Lambda} = \text{diag}\left(\frac{1}{1-2\lambda\eta_1}, \frac{1}{1-2\lambda\eta_2}, ...., \frac{1}{1-2\lambda\eta_n}\right)$.

Now we show that $S^T\bar{\Lambda}S \succeq (S^{(i)})^T\bar{\Lambda}^{(i)}S^{(i)}$ for any $i$. By Corollary 7.7.4 (a) of [22], if $P \succ 0$ and $Q \succ 0$, then $P \succeq Q$ if and only if $Q^{-1} \succeq P^{-1}$. Since we assume $0 < \lambda < \frac{1}{2\eta_1}$, it follows that $1 - 2\lambda\eta_j^{(i)} > 0$ and $1 - 2\lambda\eta_j > 0$ for any $i, j$. Thus, all diagonal elements of $\bar{\Lambda}^{(i)}$ and $\bar{\Lambda}$ are positive, implying that $(S^{(i)})^T\bar{\Lambda}^{(i)}S^{(i)} \succ 0$ and $S^T\bar{\Lambda}S \succ 0$.

Since $\left((S^{(i)})^T\bar{\Lambda}^{(i)}S^{(i)}\right)^{-1} = (S^{(i)})^T\left(I - 2\lambda\Lambda^{(i)}\right)S^{(i)} = I - 2\lambda A^{(i)}$ and $\left(S^T\bar{\Lambda}S\right)^{-1} = I - 2\lambda A$, we have

$$
\left((S^{(i)})^T\bar{\Lambda}^{(i)}S^{(i)}\right)^{-1} \succeq \left(S^T\bar{\Lambda}S\right)^{-1} \iff I - 2\lambda A^{(i)} \succeq I - 2\lambda A \iff A \succeq A^{(i)}
$$

Thus, $S^T\bar{\Lambda}S \succeq (S^{(i)})^T\bar{\Lambda}^{(i)}S^{(i)}$ holds, implying that $(\mu^i)^T S^T\bar{\Lambda}S\mu^i \geq (\mu^i)^T(S^{(i)})^T\bar{\Lambda}^{(i)}S^{(i)}\mu^i$ for any $\mu^i$. Therefore,

$$
\frac{1}{C}\,\mathbb{E}_{\pi'}\left[e^{\lambda R^{\text{true}}(W)}\right] = \prod_{i=1}^{n}\frac{\exp\left(\lambda(\mu^i)^T(S^{(i)})^T\bar{\Lambda}^{(i)}S^{(i)}\mu^i\right)}{\prod_{j=1}^{n}\left(1-2\lambda\eta_j^{(i)}\right)^{1/2}} \leq \prod_{i=1}^{n}\frac{\exp\left(\lambda(\mu^i)^T S^T\bar{\Lambda}S\mu^i\right)}{\prod_{j=1}^{n}\left(1-2\lambda\eta_j\right)^{1/2}} = \frac{1}{C}\,\mathbb{E}_{\pi}\left[e^{\lambda R^{\text{true}}(W)}\right]
$$

$\square$

## B  Further Discussion on the Convergence of Shalaeva's Bound

Another convergence result by Shalaeva el al [45] is that: For any fixed $\pi, \rho, \delta$ such that $D(\rho\,||\,\pi) < \infty$, take $\lambda = m^{1/d}$ for some constant $d > 2$, then the right hand side of Shalaeva's bound (See Section 2) converges to the left hand side as $m \to \infty$.

This result is based on the convergence of the upper bound $\mathbb{E}_{W\sim\pi}\exp\left(\frac{2\lambda^2 v_W^2}{m}\right)$: By taking $\lambda = m^{1/d}$, we have $\mathbb{E}_{W\sim\pi}\exp\left(\frac{2\lambda^2 v_W^2}{m}\right) = \mathbb{E}_{W\sim\pi}\exp\left(2m^{2/d-1}v_W^2\right)$. In this case, the following convergence statement

$$\lim_{m\to\infty} \frac{1}{\lambda}\left[ D(\rho\,\|\,\pi) + \ln\frac{1}{\delta} + \Psi_{\pi,\mathcal{D}}(\lambda,m) \right] \leq \lim_{m\to\infty} m^{-1/d}\left[ D(\rho\,\|\,\pi) + \ln\frac{1}{\delta} \right] + \lim_{m\to\infty} m^{-1/d}\ln\mathbb{E}_{W\sim\pi}\exp\left(2m^{2/d-1}v_W^2\right) = 0$$

holds if

$$\lim_{m\to\infty} m^{-1/d}\ln\mathbb{E}_{W\sim\pi}\exp\left(2m^{2/d-1}v_W^2\right) = 0 \tag{38}$$

Shalaeva et al. [45] provided only one condition $d > 2$ to ensure (38), and they did not discuss the choice of $\pi$. However, we find that $d > 2$ alone is not sufficient to guarantee convergence, since (38) does not hold for all $\pi$. A few examples are given below:

**Example B.1.** If $\pi$ is a distribution with bounded support, then (38) holds. This is because there exists a constant $G > 0$ such that $\|W\|_2 < G$. We can show that for any $\lambda > 0$, $\mathbb{E}_{W\sim\pi}\exp\left(\lambda v_W^2\right) < \infty$:

$$\mathbb{E}_{W\sim\pi}\exp\left(\lambda v_W^2\right) = \mathbb{E}_{W\sim\pi}\left[\exp\left(\lambda(\sigma_x^2\|W^*-W\|_2^2 + \sigma_e^2)^2\right)\right] \leq \mathbb{E}_{W\sim\pi}\left[\exp\left(\lambda(\sigma_x^2(\|W^*\|_2 + \|W\|_2)^2 + \sigma_e^2)^2\right)\right]$$
$$< \mathbb{E}_{W\sim\pi}\left[\exp\left(\lambda(\sigma_x^2(\|W^*\|_2 + G)^2 + \sigma_e^2)^2\right)\right] = \exp\left(\lambda(\sigma_x^2(\|W^*\|_2 + G)^2 + \sigma_e^2)^2\right) < \infty$$

Thus, when $d > 2$,

$$\lim_{m\to\infty} m^{-1/d}\ln\mathbb{E}_{W\sim\pi}\exp\left(2m^{2/d-1}v_W^2\right) \leq \lim_{m\to\infty} m^{-1/d}\ln\mathbb{E}_{W\sim\pi}\exp\left(2v_W^2\right) = 0$$

**Example B.2.** Let $\pi$ be a Gaussian distribution. We show that (38) does not hold.

We first show that $\mathbb{E}_{W\sim\pi}\exp\left(\lambda v_W^2\right) = \infty$ for any $\lambda > 0$. Denote $w = W^* - W \in \mathbb{R}^{1\times n}$ where $W \sim \pi$, then $w$ is a Gaussian random vector. Thus

$$\mathbb{E}_{W\sim\pi}\exp\left(\lambda v_W^2\right) = \mathbb{E}_{W\sim\pi}\left[\exp\left(\lambda(\sigma_x^2\|W^*-W\|_2^2 + \sigma_e^2)^2\right)\right] \geq \mathbb{E}_{W\sim\pi}\left[\exp\left(\lambda(\sigma_x^2\|W^*-W\|_2^2)^2\right)\right]$$
$$= \mathbb{E}_w\left[\exp\left(\lambda\sigma_x^4\|w\|_2^4\right)\right] = \mathbb{E}_w\left[\exp\left(\lambda\sigma_x^4(\sum_{i=1}^n w_i^2)^2\right)\right] \geq \mathbb{E}_w\left[\exp\left(\lambda\sigma_x^4 w_1^4\right)\right]$$

Here $w_1 \in \mathbb{R}$ is the first element of $w$, which is a Gaussian random variable. Suppose $w_1 \sim \mathcal{N}(\mu, \sigma^2)$, then

$$\mathbb{E}_w\left[\exp\left(\lambda\sigma_x^4 w_1^4\right)\right] = \int \exp\left(\lambda\sigma_x^4 w_1^4\right) \cdot \frac{1}{\sqrt{2\pi}\sigma}\exp\left(-\frac{(w_1-\mu)^2}{2\sigma^2}\right)\,dw_1$$
$$= \int \frac{1}{\sqrt{2\pi}\sigma}\exp\left(\lambda\sigma_x^4 w_1^4 - \frac{(w_1-\mu)^2}{2\sigma^2}\right)\,dw_1 = \infty$$

because $\lambda\sigma_x^4 w_1^4 - \frac{(w_1-\mu)^2}{2\sigma^2} \to \infty$ as $w_1 \to \infty$. Hence, $\mathbb{E}_{W\sim\pi}\exp\left(\lambda v_W^2\right) = \infty$ for any $\lambda > 0$.

If (38) holds, then by the definition of limit, for any $\epsilon > 0$, there exists a finite integer $M$ such that for any $m > M$, $m^{-1/d}\ln\mathbb{E}_{W\sim\pi}\exp\left(2m^{2/d-1}v_W^2\right) < \epsilon$. The negation of this statement is that, there exists $\epsilon > 0$ such that for any finite integer $M$, there exists $m > M$ satisfying $m^{-1/d}\ln\mathbb{E}_{W\sim\pi}\exp\left(2m^{2/d-1}v_W^2\right) \geq \epsilon$. Let $\epsilon = 1$, $M$ be any finite integer, and $m = M+1$. Then $\mathbb{E}_{W\sim\pi}\exp\left(2m^{2/d-1}v_W^2\right) = \infty$ and $1 = \epsilon \leq m^{-1/d}\ln\mathbb{E}_{W\sim\pi}\exp\left(2m^{2/d-1}v_W^2\right) = \infty$. So the negation is true, and (38) does not hold.

In summary, the validity of (38) depends on $\pi$, and a sufficient condition for convergence requires that $\pi$ be appropriately specified.

## C   Allowing Multiple Trails on $\lambda$

Finding the optimal $\lambda$ that yields the tightest bound is nontrivial. As suggested in Section 2.1.4 of [4], we approximate the optimal $\lambda$ by searching over a finite grid $\Lambda = \{\lambda_1, \lambda_2, ..., \lambda_L\}$, where each $\lambda_i > 0$ and $L$ denotes the cardinality of $\Lambda$.

$$P\left(\forall\lambda\in\Lambda, \forall\rho,\ \mathbb{E}_{W\sim\rho}[R^{\text{true}}(W)] < \mathbb{E}_{W\sim\rho}[R^{\text{emp}}(W)] + \frac{1}{\lambda}\left[D(\rho\,\|\,\pi) + \ln\frac{L}{\delta} + \Psi_{\pi,\mathcal{D}}(\lambda,m)\right]\right) \geq 1-\delta$$

This is because

$$P\left(\forall \lambda \in \Lambda, \forall \rho, \ \mathbb{E}_{W \sim \rho}[R^{\text{true}}(W)] < \mathbb{E}_{W \sim \rho}[R^{\text{emp}}(W)] + \frac{1}{\lambda}\left[D(\rho \,\|\, \pi) + \ln \frac{L}{\delta} + \Psi_{\pi, \mathcal{D}}(\lambda, m)\right]\right)$$

$$= 1 - P\left(\exists \lambda \in \Lambda, \exists \rho, \ \mathbb{E}_{W \sim \rho}[R^{\text{true}}(W)] \geq \mathbb{E}_{W \sim \rho}[R^{\text{emp}}(W)] + \frac{1}{\lambda}\left[D(\rho \,\|\, \pi) + \ln \frac{L}{\delta} + \Psi_{\pi, \mathcal{D}}(\lambda, m)\right]\right)$$

$$= 1 - P\left(\bigcup_{i=1}^{L}\left\{\exists \rho, \ \mathbb{E}_{W \sim \rho}[R^{\text{true}}(W)] \geq \mathbb{E}_{W \sim \rho}[R^{\text{emp}}(W)] + \frac{1}{\lambda_i}\left[D(\rho \,\|\, \pi) + \ln \frac{L}{\delta} + \Psi_{\pi, \mathcal{D}}(\lambda_i, m)\right]\right\}\right)$$

$$\geq 1 - \sum_{i=1}^{L} P\left(\exists \rho, \ \mathbb{E}_{W \sim \rho}[R^{\text{true}}(W)] \geq \mathbb{E}_{W \sim \rho}[R^{\text{emp}}(W)] + \frac{1}{\lambda_i}\left[D(\rho \,\|\, \pi) + \ln \frac{L}{\delta} + \Psi_{\pi, \mathcal{D}}(\lambda_i, m)\right]\right)$$

$$\geq 1 - \sum_{i=1}^{L} \frac{\delta}{L} = 1 - \delta$$

The grid search is a standard method for optimizing $\lambda$ and is also used by Dziugaite and Roy [13]. Note that $\lambda$ cannot be directly optimized via gradient descent while keeping $\rho$ and $\pi$, as this would make the optimal $\lambda$ depend on the dataset $S$, which is a random variable [4, 43]. Consequently, the optimal $\lambda$ would itself become a random variable, contradicting the assumption that it is fixed.

## D   Related Works

In statistical learning, generalization bounds commonly provide an upper bound on the generalization gap, $R^{\text{true}}(W) - R^{\text{emp}}(W)$ (or, in the two-sided case, $|R^{\text{true}}(W) - R^{\text{emp}}(W)|$), thereby estimating the true risk $R^{\text{true}}(W)$ for any given model $W$. A generalization bound is called a Probably Approximately Correct (PAC) bound if it guarantees, with probability at least $1 - \delta$, that the gap does not exceed a small $\epsilon$. It is PAC-Bayesian if, in addition, the model $W$ is treated as a random variable and a KL-divergence term, $D(\rho \,\|\, \pi)$, is introduced to allow Bayesian inference between the prior $\pi$ and posterior $\rho$ of $W$. Classic PAC bounds typically estimate the worst-case generalization gap, $\sup_W \{R^{\text{true}}(W) - R^{\text{emp}}(W)\}$ [51], which often becomes loose as the number of parameters in $W$ increases [39, 41]. In contrast, PAC-Bayes bounds estimate the expected generalization gap, $\mathbb{E}_W[R^{\text{true}}(W) - R^{\text{emp}}(W)]$, which is typically much tighter than the worst-case bound [13].

Early PAC-Bayes bounds, including those by McAllester [37], Catoni [9], Langford and Seeger [29], primarily focus on binary or bounded losses. In recent years, PAC-Bayes bounds have been extended to unbounded losses, as in Alquier et al. [5], Haddouche et al. [19], Haddouche and Guedj [18], Rodríguez-Gálvez et al. [43], Casado et al. [8]. These works typically assume that the loss follows a light-tailed or heavy-tailed distribution. However, such assumptions usually rely on oracle knowledge of the underlying data distribution; consequently, these bounds are often oracle bounds and can only be computed when the true data distribution is known.

Linear regression typically employs the squared loss, which is unbounded. Several PAC-Bayes bounds targeting the generalization gap have been proposed for linear regression. One major line of work adapts the unbounded squared loss to the framework of Alquier et al. [5], such as Germain et al. [17] and Shalaeva et al. [45], which we follow here. Other studies, such as Haddouche et al. [19], consider the $\ell_1$ loss for linear regression instead of the standard $\ell_2$ (squared) loss. To the best of our knowledge, all these bounds focus on the single-output setting and have not yet been extended to the multivariate case.

Another class of generalization bounds for linear regression targets the excess risk $R^{\text{true}}(W) - R^{\text{true}}(W^*)$, where $W^*$ denotes the minimizer of the true risk. The advantage of this formulation is that, when the tail distribution of the loss satisfies the Bernstein assumption (Definition 4.1, [4]), such bounds (including both PAC and PAC-Bayes types) typically achieve a $1/m$ convergence rate, which is faster than the $1/\sqrt{m}$ rate commonly observed for PAC-Bayes bounds targeting the generalization gap (Section 4.2, [4]). Several bounds for single-output and multivariate linear regression follow this setting, including those by Alquier and Bieu [3], Alquier [2], and Mai [34]. However, these bounds do not depend on the empirical risk, making them less practical for real-world applications.

Collaborative filtering can be viewed as a special case of matrix completion problem, i.e., predicting missing values in a matrix (Section 1.3.1.2, [1]). In recent years, LAEs have become a popular

model for collaborative filtering due to their simplicity and effectiveness. Unlike other models, LAEs have the distinctive property that their training objective, such as in (4), resembles a constrained linear regression problem where the input and target matrices are the same, highlighting their close relationship with linear regression. The earliest LAE model can be traced back to SLIM [40], followed by models such as EASE [48], EDLAE [49] and ELSA [50]. All of these models introduce a zero-diagonal constraint on the weight matrix $W$, preventing items from learning themselves and thereby avoiding overfitting toward the identity. A recent study by Moon et al. [38] shows that this zero-diagonal constraint can be slightly relaxed to a diagonal with small bounded norm, potentially improving performance.

While the generalizability of LAE models remains unexplored, related work has investigated generalizability in matrix completion for collaborative filtering, including Srebro et al. [47], Shamir et al. [46], and Foygel et al. [16]. Other studies focus on the generalizability of general matrix completion, such as Candès and Tao [7], Recht [42], Ledent et al. [31], and Ledent and Alves [30]. Our work differs from these studies in that we analyze LAEs mainly from the perspective of linear regression rather than matrix completion. In addition, Variational Autoencoders (VAEs) are another type of collaborative filtering model [32], and PAC-Bayes bounds have been developed for VAEs [10]. It should be noted that VAEs and LAEs are fundamentally different models, so the PAC-Bayes bounds for VAEs are not directly comparable to our bounds, as discussed in Appendix E.3.

Therefore, to the best of our knowledge, we propose the first PAC-Bayes bound for multivariate linear regression targeting the generalization gap, and the first PAC-Bayes bound for LAEs.

# E   Conclusions and Discussions

This paper studies the generalizability of multivariate linear regression and LAEs. We first propose a PAC-Bayes bound for multivariate linear regression under a Gaussian data assumption, extending Shalaeva's bound for single-output linear regression, and establish a sufficient condition that guarantees convergence. Next, we build the connection between multivariate linear regression and LAE models by introducing a relaxed MSE as an evaluation metric, under which LAE models can be interpreted as multivariate linear regression on bounded data, subject to a zero-diagonal constraint on weights and a hold-out constraint on the input and target data. This connection allows us to adapt our bound for multivariate linear regression to LAEs.

In practice, LAEs are typically large models evaluated on large datasets, which makes computing the tightest bound inefficient. To address this, we develop theoretical methods to improve computational efficiency. Specifically, by restricting both the prior and posterior to be Gaussian, we obtain an efficient sub-optimal bound in closed-form. We then address the computational cost imposed by the zero-diagonal constraint by establishing and computing an upper bound with reduced complexity. Experimental results demonstrate that our bound is tight and correlates strongly with practical metrics such as Recall@K and NDCG@K, suggesting that it effectively reflects the real-world performance of LAE models.

Below are the discussions of our work.

## E.1   Limitations

One limitation of our work lies in Algorithm 1, which takes $\Sigma_{hh} = \mathbb{E}_{h \sim \mathcal{M}}[hh^T]$ as input and requires it to be known, which is only possible when $\mathcal{M}$ is non-oracle. Whether $\mathcal{M}$ is non-oracle depends on how the dataset is modeled statistically. If the dataset $H$ is drawn from a meta-dataset $H^{\text{whole}}$ that is known and of fixed size, we may assume $\mathcal{M}$ to be the population distribution of this meta-dataset, thereby making $\mathcal{M}$ non-oracle. However, this assumption is rather restrictive: In most real-world scenarios, such a meta-dataset may not exist, as data are continuously collected or expand over time. In this case, $\mathcal{M}$ is typically modeled as an unknown oracle distribution to account for unseen data. Therefore, whether $\mathcal{M}$ is non-oracle ultimately depends on how the dataset is modeled, and our work shows that the bound is at least computable in restricted scenarios where datasets can be represented by a non-oracle $\mathcal{M}$.

To avoid introducing an oracle $\mathcal{M}$, one may also consider reconstructing the bound by applying linear regression to empirical PAC-Bayes bounds such as those of McAllester [37], and Langford and Seeger [29], rather than to the Alquier's oracle bound [5]. However, this approach is not feasible

because empirical PAC-Bayes bounds are typically derived under the assumption of bounded loss, whereas the loss in linear regression is unbounded since $W$ itself is unbounded in $\|y_i - Wx_i\|_F^2$.

Moreover, PAC-Bayes bounds for unbounded losses are inherently oracle, as they are derived from tail-distribution assumptions that are themselves oracle in nature. These distributions are usually characterized by a bounded exponential moment $\mathbb{E}_{X \sim \mathcal{M}}\left[e^{\lambda(\mathbb{E}_{X \sim \mathcal{M}}[X] - X)}\right]$ for some $\lambda \in \mathbb{R}$, as in the Hoeffding assumption [5] and the bounded Cumulant Generating Function (CGF) assumption [43, 19]. Assuming these quantities are bounded implicitly presumes access to oracle information about $\mathcal{M}$, which is rarely realistic in practice.

Another limitation is that the computational methods introduced in Section 5 are primarily designed for full rank or nearly full-rank LAEs. By 'nearly-full rank', we refer to matrices of rank $n-1$ formed by applying the zero-diagonal constraint to a full-rank $n \times n$ matrix. Some LAE models, however, impose low-rank constraints [49] on $W$, where a $W^{n \times n}$ of rank $k$ $(k < n)$ can be decomposed as $W = UV$ with $U^{n \times k}$ and $V^{k \times n}$. Assumption 5.1 does not hold in this case, since it requires $W$ to be a random Gaussian matrix, implying full rank [14]. Consequently, results that rely on this assumption, including Theorems 5.2 and 5.4, are not applicable to low-rank LAEs. One possible approach to adapting our bound for low-rank $W$ is to impose distributional assumptions on $U$ and $V$ so that any realization of $W = UV$ is always low-rank; however, this contradicts Assumption 5.1.

### E.2 Comparison with Mai's Excess Risk Bounds [34]

Mai proposed generalization bounds for bilinear regression (Theorem 2, [34]) and matrix completion (Theorem 3, [34]) based on excess risk. Their bilinear regression setting is defined as follows: Given two input matrices $Z \in \mathbb{R}^{p \times r}$ and $X \in \mathbb{R}^{n \times m}$, let $Y \in \mathbb{R}^{p \times m}$ be the target matrix following a distribution conditioned on $Z$ and $X$, then there exists $W^* \in \mathbb{R}^{r \times n}$ such that $Y = ZW^*X + E$, where $E \in \mathbb{R}^{p \times m}$ is a random noise matrix whose entries $E_{ij}$ has zero mean and finite variance. If taking $r = p$ and $Z = I$, it reduces to the multivariate linear regression presented in Section 2.

Given a model $W \in \mathbb{R}^{p \times m}$, let the true risk be defined as $R^{\text{true}}(W) = \frac{1}{pm}\mathbb{E}_{Y|Z,X}[\|Y - ZWX\|_F^2]$, then the excess risk can be expressed as

$$R^{\text{true}}(W) - R^{\text{true}}(W^*) = \frac{1}{pm}\|ZWX - ZW^*X\|_F^2$$

If we further suppose that $W$ is a random variable following a distribution $\hat{\rho}$, and define the expected excess risk as $\mathbb{E}_{W \sim \hat{\rho}}[R^{\text{true}}(W)] - R^{\text{true}}(W^*)$, then by Jensen's inequality we have

$$\mathbb{E}_{W \sim \hat{\rho}}[R^{\text{true}}(W)] - R^{\text{true}}(W^*) \geq \frac{1}{pm}\left\|\mathbb{E}_{W \sim \hat{\rho}}[ZWX] - ZW^*X\right\|_F^2 \tag{39}$$

Mai first derived a generalization bound for bilinear regression by establishing an upper bound on $\|\mathbb{E}_{W \sim \hat{\rho}}[ZWX] - ZW^*X\|_F^2$. They then adapted this bound to the matrix completion setting, which assumes that $k(k < pm)$ pairs in $\{((ZW^*X)_{ij}, Y_{ij})\}$ are observed, where the trained model $W$ is used to recover the remaining pairs. Further, their bound converges linearly with respect to $k$.

It is well known that if the loss satisfies Bernstein assumption (Definition 4.1, [4]), one can construct an upper bound on the excess risk (or the expected excess risk) that converges at a linear rate of $1/m$. By (39), Mai's bound is indeed an upper bound on a *lower bound* of the excess risk, which makes its linear convergence rate reasonable.

In contrast, our bounds are based on the generalization gap $R^{\text{true}}(W) - R^{\text{emp}}(W)$, rather than the excess risk. PAC-Bayes bounds on the generalization gap typically converge at a slower rate of $1/\sqrt{m}$ (Section 4.2, [4]). The Bernstein assumption, which enables linear convergence for excess risk bounds, is not applicable in the setting of the generalization gap.

### E.3 Comparison with PAC-Bayes Bounds for Variational Autoencoders [10]

Like LAEs, Variational Autoencoders (VAEs) are another class of autoencoders that have been applied to collaborative filtering [32]. Recently, PAC-Bayes bounds for VAEs were proposed by Chérief-Abdellatif et al. [10].

Although LAEs and VAEs are both autoencoders, they differ fundamentally in model architecture, learning objectives, and loss formulations, making our bound for LAEs not directly comparable to the PAC-Bayes bound for VAEs.

In detail, the architecture of VAEs [10] is:

$$\text{input } \mathbf{x} \xrightarrow{\text{encoder } q_\phi(\mathbf{z}|\mathbf{x})} \text{latent code } \mathbf{z} \xrightarrow{\text{decoder } p_\theta(\mathbf{x}|\mathbf{z})} \text{prediction} \quad \approx \quad \text{target } \mathbf{x}$$

with the loss defined as $-\mathbb{E}_{q_\phi(\mathbf{z}|\mathbf{x})}[\log p_\theta(\mathbf{x}|\mathbf{z})]$.

And the architecture of LAEs is

$$\text{input } \mathbf{x} \xrightarrow{\text{encoder and decoder } W} \text{prediction } W\mathbf{x} \quad \approx \quad \text{target } \mathbf{y}$$

with the loss defined as the relaxed MSE $\|\mathbf{y} - W\mathbf{x}\|_F^2$ in our framework, see Section 4.2.

The differences between VAEs and LAEs are summarized as follows:

**Difference in Architecture**: A VAE consists of an encoder and a decoder and explicitly involves the latent code $\mathbf{z}$. In an LAE, the model $W$ serves as both the encoder and the decoder, with the latent code being implicit. While one could decompose $W = AB$ and treat $B$ as the encoder and $A$ as the decoder, such a decomposition is uncommon in recommender systems. Most LAE recommender models do not decompose $W$ [40, 48, 49, 50]; instead, they focus on studying $W$ directly by introducing constraints such as a zero diagonal. As a result, LAEs and VAEs are generally regarded as two distinct model classes.

**Difference in Input and Target**: VAEs require the input and target to be identical. In contrast, for LAEs, the input $\mathbf{x}$ and target $\mathbf{y}$ are the same during training but differ during evaluation due to the hold-out constraint. Our bound is defined for evaluation, where $\mathbf{x}$ and $\mathbf{y}$ are indeed different. This fundamental difference in input–target configuration makes LAEs incompatible with the VAE framework.

**Difference in Loss**: The VAE loss $-\mathbb{E}_{q_\phi(\mathbf{z}|\mathbf{x})}[\log p_\theta(\mathbf{x}|\mathbf{z})]$ aims to minimize the mismatch between the encoder $q_\phi(\mathbf{z}|\mathbf{x})$ and the decoder $p_\theta(\mathbf{x}|\mathbf{z})$, while the LAE loss $\|\mathbf{y} - W\mathbf{x}\|_F^2$ aims to minimize the mismatch between the target $\mathbf{y}$ and the prediction $W\mathbf{x}$. These losses reflect fundamentally different learning objectives, so the two models cannot share the same loss function. Moreover, the LAE loss aligns more closely with a multivariate linear regression loss than with the VAE loss, which is why our bound for multivariate linear regression can be naturally extended to LAE models.

Therefore, due to the fundamental differences between VAEs and LAEs, our bound for LAEs is not directly comparable to the PAC-Bayes bound for VAEs proposed by Chérief-Abdellatif et al. [10].

### E.4  Relationship between MSE and Ranking Metrics

While minimizing MSE is not theoretically consistent with achieving optimal ranking, empirical studies have observed a strong correlation between reductions in MSE and improvements in ranking metrics [26]. Moreover, ranking metrics are typically set-based, discrete, and non-differentiable, making them difficult to optimize directly using gradient-based methods. Consequently, it has become standard practice in collaborative filtering to employ regression-style or reconstruction-based surrogate losses – most commonly the squared error – as training objectives (Section 2.6, [1]), which resemble the MSE used during evaluation. This approach underlies many successful algorithms, including SLIM [40], EASE [48], EDLAE [49], ELSA [50], and matrix factorization models [23, 27], which consistently demonstrate strong empirical performance on ranking tasks despite optimizing a non-ranking loss.

Since LAEs typically use a linear regression loss for training and ranking metrics for evaluation – and model performance is ultimately measured by the latter – we derive our generalization bound solely with respect to the evaluation metric. As discussed in Section 4.2, because ranking metrics are difficult to analyze statistically, we instead adopt MSE as the evaluation metric, which follows the form of a linear regression loss but differs from the training loss in both definition and purpose.

### E.5  Broader Impacts

This work advances the theoretical foundations of machine learning by introducing the first PAC-Bayes bound for multivariate linear regression targeting generalization gap, extending beyond single-output regression to handle multiple dependent variables simultaneously. This establishes new generalization guarantees for structured prediction, multi-task learning, and recommendation systems.

Additionally, we identify and correct a limitation in an existing PAC-Bayes proof for single-output linear regression, further strengthening the theoretical foundation of regression analysis.

Building on this, we apply our bound to LAEs in recommendation systems, delivering their first rigorous generalization analysis. Our approach accounts for key structural constraints, such as the zero-diagonal weight requirement, ensuring applicability to models like EASE and EDLAE.

Beyond theory, our work has direct practical implications for model evaluation and selection. Our bound provides a post-training diagnostic tool for assessing the generalization of any LAE model, regardless of its training process. While not directly guiding training or hyperparameter tuning, a smaller PAC-Bayes bound suggests better generalization on unseen data. Empirical results confirm that our bound remains within a reasonable multiple of the test error, offering reliable probabilistic estimates of true risk independent of training error.

Our work focuses on theoretical generalization analysis and poses no immediate ethical risks. However, recommendation systems shape content exposure and user behavior in domains like e-commerce and social media. Strengthening generalization theory alongside other recommendation criteria may help mitigate bias, enhance fairness, and improve trust in AI-driven systems.

# F  Other Supplemental Materials

## F.1  Details of Algorithm 1

Here we provide details on the computation of $\mathbb{E}_{W \sim \rho}[R^{\mathrm{emp}}(W)]$, $\mathbb{E}_{W \sim \rho}[R^{\mathrm{true}}(W)]$ and $D(\rho \| \pi)$ in Algorithm 1, which are not fully described in the main paper.

Given $\lambda$ and $\pi = \bar{\mathcal{N}}(\mathcal{U}_0, \sigma^2 I)$, the optimal $\rho = \bar{\mathcal{N}}(\mathcal{U}, \mathcal{S})$ that minimizes the right hand side of (14) is obtained by Theorem 5.2. Once $\rho$ is obtained, we can compute $\mathbb{E}_{W \sim \rho}[R^{\mathrm{emp}}(W)]$ by (24), which can be simplified as

$$\mathbb{E}_{W \sim \rho}[R^{\mathrm{emp}}(W)] = \frac{1}{m}\|Y - \mathcal{U}X\|_F^2 + \frac{n-1}{m}\|\operatorname{diag}(\mathcal{S}_{1*})^{1/2}X\|_F^2 \tag{40}$$

The $n - 1$ term in (40) is due to the zero-diagonal constraint, which enforces $\operatorname{diag}(\mathcal{S}) = 0$. Without this constraint, the term becomes $n$ instead of $n - 1$.

Similarly, by (6), $\mathbb{E}_{W \sim \rho}[R^{\mathrm{true}}(W)]$ can be expressed as

$$\mathbb{E}_{W \sim \rho}[R^{\mathrm{true}}(W)] = \|\Sigma_{xy}^T \Sigma_{xx}^{-1/2} - \mathcal{U}\Sigma_{xx}^{1/2}\|_F^2 + (n-1)\|\operatorname{diag}(\mathcal{S}_{1*})^{1/2}\Sigma_{xx}^{1/2}\|_F^2$$
$$+ \operatorname{tr}(\Sigma_{yy}) - \|\Sigma_{xy}^T \Sigma_{xx}^{-1/2}\|_F^2 \tag{41}$$

The derivation of (41) is analogous to (24) by substituting $Y$ with $\Sigma_{xy}^T \Sigma_{xx}^{-1/2}$ and $X$ with $\Sigma_{xx}^{-1/2}$.

The $D(\rho \| \pi)$ term under zero-diagonal constraint is obtained from (25) by removing the diagonal elements of $\mathcal{S}$:

$$D(\rho \| \pi) = \frac{1}{2}\left[(n^2 - n)(2\ln\sigma - 1) - \sum_{k=1}^{n}\sum_{l=1,l\neq k}^{n}\left(\ln\mathcal{S}_{kl} - \frac{\mathcal{S}_{kl}}{\sigma^2}\right) + \frac{\|\mathcal{U} - \mathcal{U}_0\|_F^2}{\sigma^2}\right] \tag{42}$$

## F.2  Dataset Description

The following table shows the details of the datasets used in the experiments.

Table 2: Dataset description

| Dataset | ML 20M | Netflix | MSD |
|---|---|---|---|
| #users ($m$) | 138493 | 480189 | 1017982 |
| #items ($n$) | 26744 | 17770 | 40000 |
| #interactions | 2000263 | 100480507 | 33687193 |

## F.3 Details of the Results in Table 1

The following table presents the detailed values of the components of the RH terms in Table 1, illustrating how the results were obtained. This information may be helpful for reproducing the experiments.

Table 3: Details of the terms of each RH in Table 1

| Models | | PAC-Bayes Bound for LAEs | | |
|---|---|---|---|---|
| | | ML 20M | Netflix | MSD |
| $\gamma = 50$ | $\lambda$ | 512 | 512 | 512 |
| | $\mathbb{E}_{W \sim \rho}[R^{\mathrm{emp}}(W)]$ | 66.99 | 90.87 | 16.58 |
| | $D(\rho \,\|\, \pi)$ | 0.28 | 0.18 | 0.0019 |
| | $\ln \mathbb{E}_{\pi}\left[e^{\lambda R^{\mathrm{true}}(W)}\right]$ | 31571.14 | 44659.37 | 8196.30 |
| $\gamma = 100$ | $\lambda$ | 512 | 512 | 512 |
| | $\mathbb{E}_{W \sim \rho}[R^{\mathrm{emp}}(W)]$ | 65.14 | 89.68 | 16.34 |
| | $D(\rho \,\|\, \pi)$ | 0.27 | 0.17 | 0.0018 |
| | $\ln \mathbb{E}_{\pi}\left[e^{\lambda R^{\mathrm{true}}(W)}\right]$ | 31102.53 | 44313.39 | 8141.72 |
| $\gamma = 200$ | $\lambda$ | 512 | 512 | 512 |
| | $\mathbb{E}_{W \sim \rho}[R^{\mathrm{emp}}(W)]$ | 63.59 | 88.57 | 16.12 |
| | $D(\rho \,\|\, \pi)$ | 0.26 | 0.17 | 0.0018 |
| | $\ln \mathbb{E}_{\pi}\left[e^{\lambda R^{\mathrm{true}}(W)}\right]$ | 30753.19 | 44014.86 | 8092.62 |
| $\gamma = 500$ | $\lambda$ | 512 | 512 | 512 |
| | $\mathbb{E}_{W \sim \rho}[R^{\mathrm{emp}}(W)]$ | 61.93 | 87.26 | 15.89 |
| | $D(\rho \,\|\, \pi)$ | 0.23 | 0.17 | 0.0016 |
| | $\ln \mathbb{E}_{\pi}\left[e^{\lambda R^{\mathrm{true}}(W)}\right]$ | 30444.39 | 43703.53 | 8044.64 |
| $\gamma = 1000$ | $\lambda$ | 512 | 512 | 512 |
| | $\mathbb{E}_{W \sim \rho}[R^{\mathrm{emp}}(W)]$ | 60.96 | 86.42 | 15.79 |
| | $D(\rho \,\|\, \pi)$ | 0.23 | 0.16 | 0.0016 |
| | $\ln \mathbb{E}_{\pi}\left[e^{\lambda R^{\mathrm{true}}(W)}\right]$ | 30310.47 | 43522.96 | 8033.10 |
| $\gamma = 2000$ | $\lambda$ | 512 | 512 | 512 |
| | $\mathbb{E}_{W \sim \rho}[R^{\mathrm{emp}}(W)]$ | 60.23 | 85.71 | 15.78 |
| | $D(\rho \,\|\, \pi)$ | 0.22 | 0.15 | 0.0015 |
| | $\ln \mathbb{E}_{\pi}\left[e^{\lambda R^{\mathrm{true}}(W)}\right]$ | 30255.43 | 43382.46 | 8052.90 |
| $\gamma = 5000$ | $\lambda$ | 512 | 512 | 512 |
| | $\mathbb{E}_{W \sim \rho}[R^{\mathrm{emp}}(W)]$ | 59.70 | 84.97 | 15.88 |
| | $D(\rho \,\|\, \pi)$ | 0.20 | 0.14 | 0.0014 |
| | $\ln \mathbb{E}_{\pi}\left[e^{\lambda R^{\mathrm{true}}(W)}\right]$ | 30308.30 | 43255.35 | 8128.97 |

