# OpenReview forum: "PAC-Bayes Bounds for Multivariate Linear Regression and Linear Autoencoders"
_NeurIPS.cc/2025/Conference — NeurIPS 2025 poster_

### Official Review · Reviewer_fRvV · 2025-06-23

**Clarity:** 1
**Significance:** 2
**Originality:** 2
**Rating:** 4
**Confidence:** 2

**Summary:**

This paper provides a novel theoretical understanding of Linear AutoEncoders (LAEs) through PAC-Bayes theory. First, the authors extend an existing result of Shalaeva et al. to the case of multivariate linear regression and explicit sufficient conditions to ensure convergence of such bounds. Then, they adapt their results to LAEs by focusing on the particular case of multivariate linear regression on bounded data. From such theoretical results, they derive a novel learning algorithm for LAEs and conclude their work with some experiments.

**Questions:**

- L. 208 what is $X_{*,j}$ in practice? The star suggest this is a theoretical quantity in some sense, is this true?
- Lemma 4.2 why highlighting the specific form of $R_{true}$ given this quantity cannot be computable in practice?
- Theorem 3.2: can you compare your result PAC-Bayes bounds existing for linear regression here (instead of the appendix)? For instance, with either [1,2] or Shalaeva's work to understand precisely what is original in your bound?
- Theorem 3.2 is obtained from the general Alquier's bound, which is often achieved by subgaussian or subexponential assumptions, which is a choice you do not make here. In recent works PAC-Bayes bounds for heavy-tailed losses (i.e. admitting finite variances) have been obtained via supermartingales argument see [3,4]). Couldn't you reach tighter results using these bounds? This would avoid the exponential moment which may ba hard to control and (potentially) computationally instable
- l. 87 the assumption on $\mathcal{D}$ is not valid throughout the whole paper, you should detail it
- l.196, isn't it the exact same statement than Theorem 3.2? What is new here?
- l. 214, 'And the PAC-Bayes bound for LAEs is formed by applying the above two constraints to (8)' what does this mean?
- l. 252 You affirm that 'neural network models typically do not admit closed-form solutions for the optimal ρ for (11).' This is actually not true if you choose to assimilate a neural net to its vectors of weights. (11) is minimized by Gibbs posteriors (see e.g. [5, Section 5.1])
- l. 266-267 Are such derivation useful in the main text?
- It seems that
- Proposition 5.3 Again, why highlighting the specific form of the exponential moment (a function of $R_{true}$) given this quantity cannot be computable in practice?
- l. 291 you say this exponential moment is much easier to compute. How can you make this bound tractable as it is supposed to be unknown on real-life experiments?


In conclusion, while some theoretical novelties seems to appear in this work (PAC-Bayes bounds for the specific case of multilinear regression without light-tailed assumptions), additional discussions and comparisons with modern methods in PAC-Bayes is severely lacking and make the paper hard to parse.  As I am not an expert in LAE, I may miss the particular relevance of the proposed results but, in my humble opinion, the paper has not been truly pedagogical about it. I believe then it needs to be enriched and rewritten before acceptance.


References
[1] Germain et al., PAC-Bayesian Theory meets Bayesian inference, 2016

[2] Haddouche et al. PAC-Bayes Unleashed: Generalisation Bounds with Unbounded Losses, 2021

[3] Haddouche et al. PAC-Bayes Generalisation Bounds for Heavy-Tailed Losses through Supermartingales; 2023

[4] Chugg et al. A unified recipe for deriving (time-uniform) PAC-Bayes bounds, 2024

[5] Catoni, A PAC-Bayesian approach to adaptive classification, 2003

**Ethical Concerns:**

["NO or VERY MINOR ethics concerns only"]

**Final Justification:**

Authors answered to all my concerns about the writing of the paper. Furthermore, they seem to have answered in a convincing manner (to me) to the questions of other reviewers, while plugging those change implies a significant rewriting, it seems it can be done in a reasonable amount of time.

**Limitations:**

Literature review is missing to understand the precise novelty of such results. Also, a strength of Theorem 3.2, which is to avoid assumptions such as subgaussianity, is not highlighted.

**Paper Formatting Concerns:**

None.

**Quality:**

2

**Strengths And Weaknesses:**

Strengths:
- The question of PAC-Bayes for LAEs is, to my knowledge, new.
- Authors avoid classical assumptions in PAC-Bayes such as subgaussianity for their specific setting

Weaknesses
- The paper is hard to read and there is little discussion about the results.
- Lit review is incomplete in the main text?
- The motivation behind the various lemma and theorem is not always clear (e.g Lemmas 4.2,4.3 Proposition 5.3)

---

> ### Author Rebuttal · Authors · 2025-07-31
>
> Thank you very much for your detailed review and thoughtful feedback regarding the clarity of our paper. We hope the rebuttal below addresses all of your concerns. Due to character limits, we may not be able to address some questions in full detail. Please do not hesitate to reach out if any points remain unclear or if further concerns arise; we sincerely welcome the opportunity to clarify and improve our work.
>
> **Questions**:
>
> > Q1: L. 208 what is $X\_{*j}$ in practice? The star suggest this is a theoretical quantity in some sense, is this true?
>
> Thanks for pointing out this symbol. $X\_{*j}$ denotes the $j$th column of $X$, which indeed has practical meaning, as discussed in Lines 108-120. $X\_{\*j}$ represents the items user $j$ have observed, while $Y\_{*j}$ represents the items treated as unseen for user $j$. An LAE model $W$ that minimizes $\|\|Y\_{\*j} - WX\_{\*j}\|\|\_F^2$ aims to learn latent relationships between the observed and unseen items. This modeling of $X\_{\*j}$ and $Y\_{\*j}$ follows a widely adopted in collaborative filtering evaluation approach called *weak generalization* [1, 2, 3]. Our Constraint 2 in Section 4.2 is formulated based on this weak generalization setting.
>
> > Q2: Lemma 4.2 why highlighting the specific form of $R^{\text{true}}$ given this quantity cannot be computable in practice?
>
> Thanks for your question. Our original claim is that (7) is computable in practice, but only when the data distribution $\mathcal{D}$ is *non-oracle*. Since $\Sigma_{xx}, \Sigma_{xy}$ and $\Sigma_{yy}$ depend on $\mathcal{D}$, a non-oracle $\mathcal{D}$ ensures these quantities are known, thereby making (7) computable.
>
> However, a non-oracle $\mathcal{D}$ is quite restrictive for modeling real-world data -- it can only represent datasets that are fixed and finite, with all training and testing samples drawn from it, which are difficult (though not impossible) to find in practice. Many real-world datasets are continuously collected or grow over time and are often modeled as samples drawn from an unknown oracle distribution $\mathcal{D}$. Our contributions show that the bound is at least computable in restricted scenarios where datasets can be modeled with non-oracle distributions.
>
> > Q3: Theorem 3.2: can you compare your result PAC-Bayes bounds existing for linear regression here (instead of the appendix)? For instance, with either [1,2] or Shalaeva's work to understand precisely what is original in your bound?
>
> Thank you for this helpful suggestion. We will revise the main text to include a more explicit comparison with Shalaeva et al's work.
>
> To clarify the relationship: Shalaeva's bound (1) applies to *single-output* linear regression (i.e., $p = 1$ in our notation), while our bound (4) extends this result to the more general *multi-output* case where $p \geq 1$. As noted in Line 148, our result strictly generalizes theirs: under appropriate parameter choices, our bound reduces exactly to Shalaeva’s.
>
> More importantly, our formulation captures scenarios where multiple output dimensions may be dependent, cases not addressed by prior single-output analyses. This extension is nontrivial, as it requires a redefinition of key variance terms and a careful treatment of the loss geometry in higher-dimensional output spaces.
>
> > Q4: Theorem 3.2 is obtained from the general Alquier's bound, which is often achieved by subgaussian or subexponential assumptions, which is a choice you do not make here. In recent works PAC-Bayes bounds for heavy-tailed losses (i.e. admitting finite variances) have been obtained via supermartingales argument see [3,4]). Couldn't you reach tighter results using these bounds?
>
> > Limitations: A strength of Theorem 3.2, which is to avoid assumptions such as subgaussianity, is not highlighted.
>
> Thank you for this insightful question and for pointing us to these important recent developments.
>
> In Alquier’s general framework, sub-Gaussian or sub-exponential assumptions are commonly used to ensure that the exponential moment $\Psi\_{\pi, \mathcal{D}}(\lambda, m) = \ln \mathbb{E}\_{\pi} \mathbb{E}\_{\mathcal{D}} \left[ e^{\lambda (R^{\text{true}}(W) - R^{\text{emp}}(W))} \right]$ is bounded, but does not guarantee the convergence, that $\Psi\_{\pi, \mathcal{D}}(\lambda, m) \to 0$ as $m \to \infty$. For instance, Germain et al.[4, A.4] showed that in linear regression, $\Psi$ is sub-gamma, but Shalaeva et al. [5] later demonstrated that this bound does not converge and proposed an alternative upper bound that is both strictly tighter and convergent.
>
> Our bound builds on the work of Shalaeva et al, using a more specific assumptions on $(\lambda, \pi)$ rather than the general sub-gamma assumption on $\Psi$. As discussed in Section 3.2, certain combinations of $(\lambda, \pi)$ yield a both sub-gamma and convergent upper bound for $\Psi$, indicating that our assumptions on $(\lambda, \pi)$ are stronger than the standard sub-gamma assumption and lead to a tighter bound.
>
> Regarding the supermartingale-based bounds, we note that they also ensure convergence at a $1/\sqrt{m})$ rate (see [6], Page 8), which is the same as our bound (See Appendix B in our paper).
>
> > Q5: l. 87 the assumption on $\mathcal{D}$ is not valid throughout the whole paper, you should detail it.
>
> Thanks for pointing out this issue. We apologize for the confusion caused by the abuse of notation $\mathcal{D}$. $\mathcal{D}$ is redefined in Assumption 3.1 and again in Assumption 4.1, and both definitions differ from the one introduced in Line 87. We will make this abuse of notation explicit in the revision.
>
> > Q6: l.196, isn't it the exact same statement than Theorem 3.2? What is new here?
>
> Please see Line 195. The definitions of $R^{\text{emp}}$ and $R^{\text{true}}$ differ here: they are derived from Assumption 4.1 (bounded data), rather than Assumption 3.1 (Gaussian data). Apart from this difference, everything else in (8) remains the same as in (4).
>
> > Q7: l. 214, 'And the PAC-Bayes bound for LAEs is formed by applying the above two constraints to (8)' what does this mean?
>
> We apply the two constraints to the loss $\|\|y - Wx\|\|_F^2$ in (8). Constraint 1 applies to $W$, and Constraint 2 applies to $(x, y)$. Together, these constraints adapt the original loss to the LAE setting.
>
> > Q8: l. 252 You affirm that 'neural network models typically do not admit closed-form solutions for the optimal $\rho$ for (11).' This is actually not true if you choose to assimilate a neural net to its vectors of weights. (11) is minimized by Gibbs posteriors (see e.g. [5, Section 5.1])
>
> Thank you for your thoughtful comment and for pointing this out. While it is true that the bound in (11) is minimized by a Gibbs posterior, the Gibbs posterior is typically a complex distribution without a closed-form density function or parameterization. Therefore, although the Gibbs posterior is well-defined in theory, it is often not analytically tractable in practice.
>
> In contrast, when choosing a Gaussian posterior (as we do in our work), we can derive an explicit closed-form solution for the optimal parameters (mean and variance) under linear model assumptions, such as in the case of linear autoencoders. While this choice may lead to a looser bound compared to the Gibbs posterior, it offers significant computational advantages, especially for large-scale datasets, by avoiding iterative approximation techniques. This reflects a trade-off between bound tightness and computational efficiency.
>
> > Q9: l. 266-267 Are such derivation useful in the main text?
>
> The derivation in Lines 266 - 267 defines the constants $B$ and $C$, which are then used in forming (13) and (14).
>
> > Q10: Proposition 5.3 Again, why highlighting the specific form of the exponential moment (a function of $R^{\text{true}}$) given this quantity cannot be computable in practice?
>
> Please see our answer to Q2. The quantity in (13) of Proposition 5.3 is computable when $\mathcal{D}$ is non-oracle. Non-oracle data distributions have been used in earlier generalization bounds, for example, the bound by Srebro et al., which we cite in Line 309.
>
> > Q11: l. 291 you say this exponential moment is much easier to compute. How can you make this bound tractable as it is supposed to be unknown on real-life experiments?
>
> Please see our answers to Q2 and Q10, in real-life experiments, if the dataset remains fixed and finite over time, and all training and testing samples are drawn from it, then it can be modeled by its population distribution, which is non-oracle and makes the bound computable.
>
> **Weaknesses:**
>
> > Lit review is incomplete in the main text?
>
> We will consider moving Literature Review from Appendix to the main text if space permits in the revision.
>
> > The motivation behind the various lemma and theorem is not always clear (e.g Lemmas 4.2,4.3 Proposition 5.3)
>
> Thank you for raising this point.
>
> Lemmas 4.2 and 4.3 provide the explicit expansion of (8) under Assumption 4.1. These expansions are essential for implementing Algorithm 1, which computes the objective in (8) efficiently. Thus, the lemmas directly support the algorithmic component of our work.
>
> Proposition 5.3 gives a closed-form expression for the quantity $\mathbb{E}\_{\pi}\left[e^{\lambda R^{\textnormal{true}}(W)}\right]$, which is used in the computational complexity analysis in Lines 276--282. It also serves as a crucial intermediate step in deriving the low-complexity upper bound in Theorem 5.4.
>
> **References:**
>
> [1] Jaewan Moon et al. It's enough: Relaxing diagonal constraints in linear autoencoders for recommendation. 2023.
>
> [2] Harald Steck. Embarrassingly shallow autoencoders for sparse data. 2019.
>
> [3] Xiang Wang et al. Neural graph collaborative filtering. 2019.
>
> [4] Germain et al. PAC-Bayesian Theory meets Bayesian inference. 2016.
>
> [5] Shalaevaet al. Improved PAC-Bayesian bounds for linear regression. 2020.
>
> [6] Haddouche et al. PAC-Bayes Generalisation Bounds for Heavy-Tailed Losses through Supermartingales. 2023.

---

> > ### Comment · Reviewer_fRvV · 2025-08-04
> >
> > Thank you for your careful reply. In the light of your rebuttal, I believe that updating this work with all requested changes from all reviewers would be enough to secure acceptance. I am thus updating my score.

---

> > > ### Author Response · Authors · 2025-08-04
> > > **Thanks for your response**
> > >
> > > Thank you very much for your thoughtful and detailed reviews. We sincerely appreciate your constructive feedback and are grateful for your decision to raise the score.

---

### Official Review · Reviewer_z6h4 · 2025-06-26

**Clarity:** 3
**Significance:** 3
**Originality:** 3
**Rating:** 5
**Confidence:** 2

**Summary:**

The paper extends PAC-Bayes theory to multivariate linear regression, deriving a novel generalization bound that remains tight for full-rank models under unbounded loss. It adapts this bound to Linear Autoencoders by framing them as constrained multivariate regression on bounded data, addressing challenges like zero-diagonal weight constraints and input-target dependencies. Experiments demonstrate tightness and correlation with recommendation metrics like Recall@K and NDCG@K.

Strengths:

First PAC-Bayes bound for multivariate regression and LAEs, filling a theoretical gap.
Rigorous convergence analysis with sufficient conditions.
Computationally efficient methods for bound optimization.
Strong empirical validation on real-world datasets.

Limitations:
The author has elaborately described his framework and has carried out numerous theoretical studies and experiments to verify its effectiveness.
Due to the large number of formulas and symbols, some parts may appear rather messy (for example, the lower half of the second page). It is hoped that the author can adjust the format and structure of the paper to make it more aesthetically pleasing and easy to read.

**Questions:**

The author has elaborately described his framework and has carried out numerous theoretical studies and experiments to verify its effectiveness.
Due to the large number of formulas and symbols, some parts may appear rather messy (for example, the lower half of the second page). It is hoped that the author can adjust the format and structure of the paper to make it more aesthetically pleasing and easy to read.

**Ethical Concerns:**

["NO or VERY MINOR ethics concerns only"]

**Final Justification:**

We are satisfied that the author's response has resolved our issues. Since our initial assessment was already positive, we have decided to maintain this score.

**Limitations:**

The author has elaborately described his framework and has carried out numerous theoretical studies and experiments to verify its effectiveness.
Due to the large number of formulas and symbols, some parts may appear rather messy (for example, the lower half of the second page). It is hoped that the author can adjust the format and structure of the paper to make it more aesthetically pleasing and easy to read.

**Quality:**

3

**Strengths And Weaknesses:**

This is the first PAC-Bayes bound for multivariate regression and LAEs, and fill a theoretical gap.
The rigorous convergence analysis with sufficient conditions shows the effectiveness of the framework.
The author has proved it is a computationally efficient methods for bound optimization.
The framework has a strong empirical validation on real-world datasets.

---

> ### Author Rebuttal · Authors · 2025-07-30
>
> Thank you very much for your insightful reviews and constructive suggestions.
>
> We will incorporate your suggestions in the revised version of the paper, including adjustments to the format and structure, particularly in the lower half of the second page as noted. We believe these improvements will enhance the clarity and presentation of our work.

---

### Official Review · Reviewer_dPHG · 2025-07-03

**Clarity:** 3
**Significance:** 2
**Originality:** 3
**Rating:** 4
**Confidence:** 3

**Summary:**

This paper develops a PAC-Bayes generalization bound tailored for multivariate linear regression. The authors then derive a sufficient conditions to guarantee its theoretical convergence under certain assumption (stated in Thm 3.3), e.g. Gaussian data assumption. They then adapt this framework to handle bounded data distributions and then specifically apply it to analyze Linear Autoencoders (LAEs). After that, under Gaussian data assumption, the authors presents a practical computational method for this LAE-specific bound. At the end, experimental results on validating the tightness of the proved bounds are provided.

**Questions:**

My question will be mainly about the assumptions on input data. I think it will help greatly if the paper can provide more detailed justifications on those assumptions, e.g., maybe they are not common, but they are important and indeed appearing in XX by evidence/literature XX.

My next question might be helpful (hopefully) for enriching the paper or future direction, if possible. Can the value of the PAC-Bayes bound after a bunch of update steps during the training process help adaptively improve the training process? Is it possible?

**Ethical Concerns:**

["NO or VERY MINOR ethics concerns only"]

**Final Justification:**

I am satisfied with the authors' responses to our questions. Since my initial rating was already positive and I still find the data assumption to be not general enough, I will maintain my score.

**Limitations:**

yes

**Quality:**

3

**Strengths And Weaknesses:**

Strengths:
(1) The paper provides the first PAC-Bayes bound for LAEs.
(2) The paper derives a sufficient condition for convergence in the case of Gaussian data.
(3) The paper also fixes the issues in Shalaeva's bound and proves the claimed bound in Shalaeva's work.

Weaknesses:
(1) The calculation of PAC-Bayes bound requires the data distribution to be known.
(2) A couple of the results requires the Gaussian data assumption.

---

> ### Author Rebuttal · Authors · 2025-07-31
>
> Thank you for your insightful and constructive comments. We hope our rebuttal addresses your concerns.
>
> > My question will be mainly about the assumptions on input data. I think it will help greatly if the paper can provide more detailed justifications on those assumptions, e.g., maybe they are not common, but they are important and indeed appearing in XX by evidence/literature XX.
>
> Thanks for highlighting this important question. The assumptions made on the input and target matrices $X$ and $Y$, as described in Lines 108–120 and 201–208, follow a standard evaluation protocol in collaborative filtering known as \textit{weak generalization} [1, 2]. Specifically, the user–item interaction matrix $R$ is binarized and then randomly split into disjoint subsets to create the input matrix $X$ and the held-out target matrix $Y$, where only the 1s (positive interactions) are split.
>
> This setup is widely adopted in recommender system literature to evaluate generalization performance on unseen interactions and to avoid data leakage between training and testing. We follow this protocol as used in prior work such as [1, 2, 3]. Thank you for raising this point, we will include these references and a clearer explanation of the weak generalization setup in the revised manuscript.
>
> > My next question might be helpful (hopefully) for enriching the paper or future direction, if possible. Can the value of the PAC-Bayes bound after a bunch of update steps during the training process help adaptively improve the training process? Is it possible?
>
> Thank you very much for this insightful and forward-looking question. We sincerely appreciate your suggestion, as it touches on a promising direction that goes beyond the current scope of our work and could meaningfully enrich future research.
>
> To clarify, in the current version of our paper, the PAC-Bayesian bound is used solely for *evaluation* purposes. As described in Lines 201–202, the LAE bound measures the generalization gap between the true and empirical risks on a held-out test set. The model $W$ used in this bound can originate from any training method and is not required to minimize the bound itself.
>
> That said, we observe in Table 1 that the bound correlates strongly with practical performance metrics such as Recall@K and NDCG@K. This empirical observation suggests that the bound could potentially serve as a surrogate objective during training. Minimizing the bound might lead to models with stronger ranking performance -- thus bridging the gap between theory and practical evaluation.
>
> Moreover, as established in Theorem 5.2, our bound admits a closed-form expression, which enables efficient computation and avoids the need for complex iterative updates. This adds to its appeal as a candidate training objective.
>
> Again, we are grateful for your suggestion, and we agree that incorporating the PAC-Bayesian bound into the training loop presents an exciting future direction. We will highlight this possibility more explicitly in our revision and hope to explore it further in future work.
>
> **References:**
>
> [1] Jaewan Moon et al. It's enough: Relaxing diagonal constraints in linear autoencoders for recommendation. SIGIR, 2023.
>
> [2] Harald Steck. Embarrassingly shallow autoencoders for sparse data. WWW, 2019.
>
> [3] Xiang Wang et al. Neural graph collaborative filtering. SIGIR, 2019.

---

> > ### Comment · Reviewer_dPHG · 2025-08-05
> > **Response to Authors' Rebuttal**
> >
> > Thanks for the detailed response. They resolved my question. I will maintain my current rating.

---

> > > ### Author Response · Authors · 2025-08-07
> > > **Thanks for your comment!**
> > >
> > > We appreciate your time and effort to review our rebuttal. We particularly thank you for your insightful and constructive comments on the data assumption and  on the suggestion to incorporate the PAC-Bayesian bound into the training loop. We will add those important points into the revision.

---

### Official Review · Reviewer_7BK6 · 2025-07-03

**Clarity:** 3
**Significance:** 2
**Originality:** 3
**Rating:** 4
**Confidence:** 2

**Summary:**

This work develops PAC-Bayes generalization bound for multivariate linear regression and extends that bound to the Linear AutoEncoders (LAEs) as a constrained form of multivariate linear regression under the bounded data assumption. To make the theory practical, an algorithm for computing the bound is developed. The bound is further applied to Linear Autoencoder (LAE) recommender systems using the EASE model for empirical validation. Experiments conducted on standard recommendation datasets confirm the bound's effectiveness in a practical setting.

**Questions:**

Related to weakness 2: Why do we see correlation in the used loss  and the ranking metrics, given that optimal loss may not be the best outcome for ranking?

**Ethical Concerns:**

["NO or VERY MINOR ethics concerns only"]

**Final Justification:**

The authors have provided clarifications related to squared loss vs. ranking metrics (Weakness 2 in the review). I will maintain my current rating.

**Limitations:**

yes

**Paper Formatting Concerns:**

N/A -- Paper formatting looks correct.

**Quality:**

3

**Strengths And Weaknesses:**

**Strengths**

- Bounds for a more general case: The paper provides a nice theoretical extension of PAC-Bayes bounds to multivariate linear regression case, while also formalizing and clarifying the the established convergence proof conditions from Shalaeva et al.
- Extension to Linear AutoEncoder: The paper does a good job of linking its theory to Linear Autoencoders (LAEs) in recommender systems. The derived multivariate bound is adjusted to account for the specific case of LAEs, like the zero-diagonal constraint on the weight matrix and how the input/target data is structured in implicit feedback scenarios (Section 4.2).
- The paper also provides an algorithm to compute the bound in practice. A neat closed-form solution for the optimal posterior is also provided under the Gaussian data assumption (Theorem 5.2). The theoretical results are backed with real-world tests in Table 1, which highlight the tightness of the bound.

**Weaknesses**

- Limited applicability to low-rank scenario: The practical computation methods, particularly the closed-form solutions in Section 5, are derived under the assumption that the weight matrix W follows a multivariate Gaussian distribution, which implies it is full-rank. This framework does not directly apply to low-rank structure: W = UV^T. Such low-rank models are common in recommendation systems, e.g. in collaborative filtering and matrix completion. Adapting the theory to impose distributions on the factors U and V would break the Gaussian assumption for W, rendering the current computational methods inapplicable. This limits the scope of the paper's impact.
- Mismatch Between Theoretical Loss and Recall/NDCG metrics: The commonly used metrics in recommendation systems (Recall/NDCG) are set-based ranking metrics (e.g. computed as Recall@K and NDCG@k for a set of k predictions). It is not clear to me why the chosen loss ($(||Y - WX||_F^2)$) in the analysis is correlated to these metrics. A model that is optimal under squared error is not guaranteed to be optimal for ranking.

---

> ### Author Rebuttal · Authors · 2025-07-30
>
> Thank you for your insightful feedback and comments. We hope the rebuttal below fully addresses your concerns, and we will incorporate this important discussion into our paper.
>
> > Weakness 2: Mismatch Between Theoretical Loss and Recall/NDCG metrics: The commonly used metrics in recommendation systems (Recall/NDCG) are set-based ranking metrics (e.g. computed as Recall@K and NDCG@k for a set of k predictions). It is not clear to me why the chosen loss ($\|\|Y - WX\|\|\_F^2$) in the analysis is correlated to these metrics. A model that is optimal under squared error is not guaranteed to be optimal for ranking.
>
> > Related to weakness 2: Why do we see correlation in the used loss and the ranking metrics, given that optimal loss may not be the best outcome for ranking?
>
> Thank you for raising this important point regarding the potential mismatch between our theoretical loss function and the evaluation metrics used in recommender systems. We hope the explanation below clarifies the rationale and empirical justification for our use of the squared loss $\|\|Y - WX\|\|\_F^2$, and its observed correlation with ranking-based metrics such as Recall@K and NDCG@K.
>
> First, it is well understood that ranking metrics, particularly Recall@K and NDCG@K in implicit feedback settings, are set-based, discrete, and non-differentiable. As such, they are difficult to optimize directly using gradient-based methods. Consequently, it has become standard practice in collaborative filtering and recommender system research to employ regression-style or reconstruction-based surrogate losses, most notably the squared error $\|\|Y - WX\|\|_F^2$, as training objectives. This approach underlies many successful algorithms, including SLIM [1], EASE [2], EDLAE [3], and matrix factorization models, which consistently demonstrate strong empirical performance on ranking tasks despite optimizing a non-ranking loss.
>
> Second, while minimizing squared loss does not theoretically guarantee optimal ranking, numerous empirical studies (including our own) observe a strong correlation between improvements in squared error and gains in Recall/NDCG. This is especially evident in *latent linear models*, where $WX$ serves as a real-valued scoring function over items. Since ranking metrics (e.g., NDCG) depend only on the relative order of scores, and squared loss penalizes deviations in score magnitudes, optimizing squared error often leads to improved ranking performance, even if this relationship is not theoretically guaranteed.
>
> Third, we believe this correlation is further reinforced in our setting by the *LAE-specific constraints* imposed on both the data and the model, as described in Section 4.2. These constraints, such as the train/test split structure and zero-diagonal regularization, are applied consistently in our bounds and in the evaluation using ranking metrics. This constraint alignment encodes *shared information* about what constitutes a good model, promoting consistency between the evaluation signals. That is, although the loss and ranking metrics target different objectives, the shared modeling constraints lead both to emphasize similar aspects of model quality.
>
> That said, these metrics do not share all information. For example, squared loss evaluates absolute prediction accuracy, while Recall and NDCG focus on the relative ranking of positive items. As a result, the correlation is not perfect. This explains the observed mismatch: the model that minimizes squared loss may not be the same as the one that maximizes Recall@K or NDCG@K, and vice versa.
>
> This effect is visible in our experiments. For example, in Table 1 on the MSD dataset, the model achieving the best Recall@50 and NDCG@100 occurs at $\gamma = 500$, while the lowest LH and RH values are attained at $\gamma = 1000$. This highlights that the loss-optimal and ranking-optimal models may differ. Nevertheless, we observe a general trend: lower LH/RH tends to correspond to better ranking performance, even if the relationship is not strictly monotonic.
>
> Finally, we emphasize that our theoretical focus is on deriving a PAC-Bayesian generalization bound for the squared error—an analytically tractable and widely used surrogate loss. While deriving bounds directly for Recall or NDCG would be desirable, such metrics are discontinuous, non-convex, and intractable for theoretical analysis. Our work thus offers principled insight into a practical surrogate loss function, bridging the gap between theoretical guarantees and empirical evaluation.
>
> In summary, although squared loss and ranking metrics optimize different objectives, their empirical alignment is well-documented and widely accepted in the literature. Our analysis provides theoretical justification for a surrogate loss function that is tractable, interpretable, and highly relevant to real-world recommendation tasks.
>
> **References:**
>
> [1] Xia Ning and George Karypis. Slim: Sparse linear methods for top-N recommender systems. ICDM, 2011.
>
> [2] Harald Steck. Embarrassingly shallow autoencoders for sparse data. WWW, 2019.
>
> [3] Harald Steck. Autoencoders that don’t overfit towards the identity. NIPS, 2020.

---

> > ### Comment · Reviewer_7BK6 · 2025-08-07
> >
> > Thank you for taking the time to answer my question and providing clarifications related to squared loss vs. ranking metrics. I would encourage the authors to discuss these insights in the final version of their paper. I will maintain my current rating.

---

> > > ### Author Response · Authors · 2025-08-07
> > > **Thanks for your comment**
> > >
> > > Thank you for your thoughtful feedback and for taking the time to engage with our work. We appreciate your comments regarding the distinction between squared loss and ranking metrics, and we will be sure to incorporate a more detailed discussion of these insights in the final version of the paper. Thank you again for your constructive review.

---

### Official Review · Reviewer_npXJ · 2025-07-08

**Clarity:** 2
**Significance:** 1
**Originality:** 1
**Rating:** 4
**Confidence:** 4

**Summary:**

The authors develop PAC-Bayes bounds for
1) multivariate linear regression problems (that is, when the output variable Y is a vector),
2) linear autoencoders (LAE).

**Questions:**

1) Please compare the results to the existing works on multivariate regression and LAE/VAE's.
2) Explain what are the improvements over these results.

**Ethical Concerns:**

["NO or VERY MINOR ethics concerns only"]

**Final Justification:**

The authors included a fair discussion with the existing papers on the topic. As a consequence, the contributions of the paper are more clearly visible now. I think this is a reasonably good contribution to the literature on generalization bounds for VAE/LAEs and as such, I think it can be published in NeurIPS.

**Limitations:**

Yes.

**Paper Formatting Concerns:**

No concerns.

**Quality:**

2

**Strengths And Weaknesses:**

General comments:
- the paper seems technically correct, and is overall well written,
- but while authors know the basic PAC-Bayes techniques, they ignore the existing literature on PAC-Bayes bounds for multivariate regression and for VAEs.

1) Multivariate regression:
Covered as a special case of https://www.mdpi.com/1099-4300/25/2/333 (take Z= Identity),
Also included in https://arxiv.org/abs/2206.08619
One observation: the rates in the above references are in 1/n (n = sample size), while all rates derived in this submission cannot be better than 1/sqrt(n).

2) LAEs:
They should compare with existing bound on VAEs:
https://proceedings.mlr.press/v151/cherief-abdellatif22a.html
https://proceedings.neurips.cc/paper_files/paper/2023/hash/b29500824d22ee9bbd25e4cd97c49b55-Abstract-Conference.html
It's not OK not to cite these papers, and the results stated here cannot be assessed if there is no rigorous comparison with these existing works.

Minor comments:
- the claim that Alquier et al.'s 2016 bound cannot be used in practice because Psi depends on the unknown data-generating process is a little weird, as the paper provides many examples where we can upper bound Psi by distribution independent quantities.

---

> ### Author Rebuttal · Authors · 2025-07-30
>
> Thank you for your thoughtful and constructive feedback. We especially appreciate your comments regarding the existing literature on PAC-Bayes bounds in the context of multivariate regression and variational autoencoders (VAEs). These works are indeed highly relevant, and we will ensure they are appropriately cited and discussed in our revised manuscript.
>
> In the rebuttal below, we clarify how our contributions relate to these prior efforts. While there are conceptual connections, the prior works address different problem settings and rely on assumptions or formulations that are not directly applicable to our framework. Through this comparison, we highlight the novel technical aspects and the distinct contributions of our approach; and we hope this adequately addresses your concerns regarding the positioning and originality of our work within the broader landscape of existing research.
>
> > Multivariate regression: Covered as a special case of [2] (take Z= Identity), Also included in [3]. One observation: the rates in the above references are in 1/n (n = sample size), while all rates derived in this submission cannot be better than 1/sqrt(n).
>
> Thank you for highlighting this relevant literature. Mai's bound [2, 3] indeed address the multivariate linear regression setting and use Bayesian techniques in deriving the bound.
>
> The focus and formulation of their bounds differ fundamentally from ours: Mai's bound is for the excess risk $R^{\text{true}}(W) - R^{\text{true}}(W^\*)$, the difference of true risks between an arbitrary model $W$ and the optimal model $W^\*$ -- such bounds are often referred to as ``non Bayesian'' PAC bounds and typically achieve a $1/n$ rate under the Bernstein assumptions [1, Section 4.2; 10]. Our bound is for the generalization gap $R^{\text{true}}(W) - R^{\text{emp}}(W)$, the difference between the true and empirical risk of the same model $W$ -- such bounds are standard PAC-Bayes bounds and typically have a $1/\sqrt{n}$ rate [1, Section 4.2]. These two types of bounds estimate different quantities and are therefore not directly comparable.
>
> In detail, we explain how Mai's bound satisfies the Bernstein assumption. By [1, Definition 4.1], let $l(W)$ denote the loss function with the weights $W$ and $S$ be the dataset, we say the *Bernstein assumption* is satisfied if there exists a constant $K$ such that
> \begin{equation*}
> \mathbb{E}\_{S}[(l(W) - l(W^\*))^2] \le K(R^{\text{true}}(W) - R^{\text{true}}(W^\*))
> \end{equation*}
>
> In the proof of Lemma 3 of Mai's bound [3], they define $l(M) = \sum\_{i=1}^n (Y\_i - \prod\_C(XM)\_{\mathcal{I}\_i})^2$ and $l(M^\*) = \sum\_{i=1}^n (Y\_i - (XM^\*)\_{\mathcal{I}\_i})^2$, and showed that under Assumption 1 and 2 in [3], there exists constant $C\_1$ such that
> \begin{equation*}
> \mathbb{E}\_{S}[(l(W) - l(W^\*))^2] \le nC\_1(R^{\text{true}}(W) - R^{\text{true}}(W^\*))
> \end{equation*}
> which implies the Bernstein assumption with $K = nC\_1$.
>
> However, the Bernstein assumption is not applicable to our bound because it only applies to the excess risk $R^{\text{true}}(W) - R^{\text{true}}(W^\*)$, whereas we focus on the generalization gap $R^{\text{true}}(W) - R^{\text{emp}}(W)$. Therefore, the rate of our bound cannot be improved in the same way as Mai's bound.
>
> > LAEs: They should compare with existing bound on VAEs [4]. It's not OK not to cite these papers, and the results stated here cannot be assessed if there is no rigorous comparison with these existing works.
>
> Thank you for pointing out this related line of work. We will certainly cite [4] in our revised manuscript and expand our literature review to include a discussion of its relevance. However, we would like to clarify that the PAC-Bayesian bounds developed for VAEs in [4] are not directly comparable to our bounds for LAE models, due to fundamental differences in model architecture, learning objectives, and loss formulations.
>
> To make these distinctions precise, we summarize the key differences below:
>
> The architecture of VAE [4] is:
> \begin{equation*}
> \text{input}\\;\mathbf{x} \xrightarrow[]{\text{encoder}\\; q\_{\phi}(\mathbf{z}|\mathbf{x})} \text{latent code}\\;\mathbf{z} \xrightarrow[]{\text{decoder}\\; p\_{\theta}(\mathbf{x}|\mathbf{z})} \text{prediction} \quad\approx\quad \text{target} \\;\mathbf{x}
> \end{equation*}
> with the loss defined as $-\mathbb{E}\_{q\_{\phi}(\mathbf{z}|\mathbf{x})}[\log p\_{\theta}(\mathbf{x}|\mathbf{z})]$.
>
> The architecture of LAE is
> \begin{equation*}
> \text{input} \\;\mathbf{x} \xrightarrow[]{\text{encoder and decoder}\\; W} \text{prediction}\\; W\mathbf{x} \quad\approx\quad \text{target}\\;\mathbf{y}
> \end{equation*}
> with the loss defined as $\|\|\mathbf{y} - W\mathbf{x}\|\|\_F^2$ in our framework.
>
> The differences between VAE and LAE are as follows:
>
> **Difference in Architecture**: VAE consists of an encoder and a decoder and explicitly involves the latent code $\mathbf{z}$. In LAE, the model $W$ serves as both the encoder and the decoder, with the latent code being implicit. While one could decompose $W = AB$ and treat $B$ as the encoder and $A$ as the decoder, such a decomposition is uncommon in recommender systems. Most LAE recommender models do not decompose $W$ [6, 8, 9]; instead, they focus on studying $W$ directly by introducing constraints such as a zero diagonal. As a result, LAE and VAE are generally regarded as two distinct models.
>
> **Difference in Input and Target**: VAE requires the input and target to be the same. In contrast, the input $\mathbf{x}$ and target $\mathbf{y}$ in LAE are different. This difference stems from the data splitting strategy described in Lines 201 – 208, known as *weak generalization*, a widely used evaluation method in recommender systems [5, 6, 7]. This fundamental difference in input–target configuration makes LAE not adaptable to VAE.
>
> **Difference in Loss**: The VAE loss $-\mathbb{E}\_{q_{\phi}(\mathbf{z}|\mathbf{x})}[\log p\_{\theta}(\mathbf{x}|\mathbf{z})]$ aims to minimize the mismatch between the encoder $q\_{\phi}(\mathbf{z}|\mathbf{x})$ and the decoder $p\_{\theta}(\mathbf{x}|\mathbf{z})$, while the LAE loss $\|\|\mathbf{y} - W\mathbf{x}\|\|\_F^2$ aims to minimize the mismatch between the target $\mathbf{y}$ and the prediction $W\mathbf{x}$. These different losses reflect fundamentally different learning objectives, meaning the two models cannot share the same loss function. Moreover, the LAE loss is less similar to the VAE loss and is instead more closely related to a multivariate linear regression loss, which is why our bound for multivariate linear regression can be naturally extended to LAE models.
>
> To sum, VAEs and LAEs serve fundamentally different modeling purposes. Their loss functions and assumptions are not aligned. Consequently, their PAC-Bayesian analyses address incompatible questions. We will include a detailed discussion of [4] in the revised manuscript, along with a clear articulation of these distinctions.
>
> > The claim that Alquier et al.'s 2016 bound cannot be used in practice because Psi depends on the unknown data-generating process is a little weird, as the paper provides many examples where we can upper bound Psi by distribution independent quantities.
>
> Thank you for this thoughtful question. We appreciate the opportunity to clarify our claim regarding the practical applicability of Alquier et al.'s bound.
>
> As shown in Line 264 of our manuscript, the upper bound of $\Psi$ used in our analysis is $ \ln \mathbb{E}\_{\pi}\left[ e^{\lambda R^{\text{true}}(W)} \right]$,
> where $R^{\text{true}}(W)$ denotes the true risk and depends on the data distribution $\mathcal{D}$. Therefore, it cannot be computed unless $\mathcal{D}$ is known, i.e., a non-oracle distribution.
>
> In our experimental setting, we work with a fixed dataset and treat the empirical distribution as the population distribution. Under this assumption, $\mathcal{D}$ is non-oracle, and the bound is computable. Therefore, Alquier’s bound is applicable in such *controlled or retrospective scenarios*, where the entire dataset is observed and finite.
>
> However, in more general and realistic settings, such as online or streaming learning, data is collected continuously, and its underlying distribution remains unknown. In these scenarios, it is more appropriate to model the data as being drawn from an unknown oracle distribution $\mathcal{D}$, rendering $\Psi$ uncomputable and the bound inapplicable.
>
> To be clear, we do *not* claim that Alquier’s bound cannot be used in practice. Rather, as discussed in Lines 302--309 and 638--646, we argue that its applicability is limited to specific cases where the data distribution can be assumed known or accurately approximated. We will revise the manuscript to better reflect this nuance and avoid any potential overstatement of our critique.
>
> **References:**
>
> [1] Alquier et al. User-friendly introduction to PAC-Bayes bounds. Foundations and Trends in Machine Learning, 2024.
>
> [2] The Tien Mai. From bilinear regression to inductive matrix completion: a quasi-Bayesian analysis. Entropy, 2023.
>
> [3] The Tien Mai, Pierre Alquier. Optimal quasi-Bayesian reduced rank regression with incomplete response. arXiv:2206.08619, 2022.
>
> [4] Chérief-Abdellatif et al. On PAC-Bayesian reconstruction guarantees for VAEs. AISTATS, 2022.
>
> [5] Jaewan Moon et al. It's enough: Relaxing diagonal constraints in linear autoencoders for recommendation. SIGIR, 2023.
>
> [6] Harald Steck. Embarrassingly shallow autoencoders for sparse data. WWW, 2019.
>
> [7] Xiang Wang et al. Neural graph collaborative filtering. SIGIR, 2019.
>
> [8] Harald Steck. Autoencoders that don’t overfit towards the identity. NIPS, 2020.
>
> [9] Vojtech Vancura et al. Scalable linear shallow autoencoder for collaborative filtering. RecSys, 2022.
>
> [10] Peter Bartlett et al. Convexity, classification, and risk bounds. Journal of the American Statistical Association, 2006.

---

> > ### Comment · Reviewer_npXJ · 2025-08-02
> > **Thank you for the thorough discussion**
> >
> > I thank the authors for taking my comments seriously and including a thorough discussion of the existing literature. The contributions are more clear now. Including these comments (as well as the corrections requested by the other reviewers) will make the paper above the bar in my opinion. I will thus update my score.
> >
> > Just one minor comment on the 1/qsrt(n) vs 1/n rate in the factorization model. I agree that the result in Mai (2023) is an excess risk bound, not directly comparable to a generalization bound. The problem is that, minimizing a generalization bound will lead to estimators with excess risk bounds in 1/sqrt(n), while the approach in Mai (2023) leads to an excess risk bound in 1/n. In other words, I don't claim that the bound proven by the authors is bad. My claim is rather that, in this setting, to minimize a generalization bound might not be the right approach as it will asymptotically lead to suboptimal excess risk. More precisely, the estimator in Mai (2023) still minimizes a bound, that is seen in Equation (A6) page 18 in the paper. This bound is sadly not a generalization bound (it depends on M* in the RHS) but to minimize it will ultimately give better predictions... As long as this is mentioned by the authors, I'm fine with it.

---

> ### Author Response · Authors · 2025-08-03
> **Thank you for your constructive feedback**
>
> Thank you very much for your thoughtful and constructive comments. We sincerely appreciate your careful reading of the paper and your positive assessment of our responses. We are especially grateful that you found the updated discussion of the existing literature helpful in clarifying the contributions.
>
> Regarding your final comment on the distinction between generalization and excess risk bounds, we appreciate the clarification. As you noted, while our PAC-Bayesian bound controls the generalization gap, minimizing it leads to estimators converging at a slower $1/\sqrt{n}$ rate, as long as the empirical risk is contained in the bound. In contrast, the estimator in Mai (2023) achieves a faster $1/n$ excess risk rate by discarding the empirical risk and introducing Bernstein assumption, resulting in a pure oracle bound [1, Remark 4.2] that differs fundamentally from ours.
>
> We agree that this is an important conceptual distinction and will explicitly mention it in the revised version to clarify the trade-off between the convergence rate and oracular nature of the bound.
>
> Once again, thank you for your insightful feedback and for helping us improve the clarity and scope of the paper.
>
> Reference:
>
> [1] Alquier et al. User-friendly introduction to PAC-Bayes bounds. Foundations and Trends in Machine Learning, 2024.

---

### Note · Authors · 2025-08-13

Dear NeurlIPS 2025 Conference Reviewers, Program Chairs, Senior Area Chairs, and Area Chairs,

We sincerely thank all reviewers for their valuable insights and constructive feedback, as well as for their thoughtful engagement during rebuttal stage. To the best of our knowledge, we present the first PAC-Bayes bound for multivariate linear regression targeting the generalization gap and establish sufficient conditions for its convergence, filling an important theoretical gap. We further extend this bound to Linear Autoencoders (LAEs) by introducing model-specific assumptions on the squared loss, propose the first PAC-Bayes bound for LAEs, develop efficient computational methods to evaluate it on large real-world datasets, and empirically show that it is tight and well-correlated with performance measured by Recall and NDCG.

In the initial review phase, we appreciate the reviewers’ recognition of our strengths, including the novelty of our contributions (Reviewers fRvV, z6h4, dPHG, and 7BK6), rigorous theoretical analysis (Reviewers z6h4, 7BK6 and npXJ), computational efficiency (Reviewers z6h4 and 7BK6), and strong empirical validation (Reviewers z6h4 and 7BK6).

During the rebuttal phase, we systematically addressed the reviewers' initial concerns and received strong recognition of our efforts. Specifically, we clarified to Reviewer npXJ that our bounds differ fundamentally from both Mai’s excess risk bound and the VAE bound, leading to their recognition of the significance of our contributions; addressed clarity and motivation concerns raised by Reviewer fRvV; acknowledged the formatting and structural issues noted by Reviewer z6h4; clarified the recommender data assumptions and potential applications of the bound for Reviewer dPHG; and provided detailed explanations regarding the mismatch between theoretical loss and Recall/NDCG metrics for Reviewer 7BK6. We also agree with Reviewer 7BK6 that our Gaussian-assumption-based computational method has limited applicability to low-rank LAE models. This limitation is discussed in Lines 656-665 of our paper, where we suggest a potential solution of replacing the Gaussian assumption with a low-rank distribution assumption to reform the computational process. Aside from this limitation, we have thoroughly addressed all other major concerns, and we will incorporate these discussions into the revised version.

We appreciate your thoughtful consideration.

Best Regards,

Authors

---

### Decision · Program_Chairs · 2025-09-17

**Decision:**

Accept (poster)

**Comment:**

### Summary of scientific claims and findings

The paper develops the first PAC-Bayes generalization bound for multivariate linear regression targeting the generalization gap, with sufficient convergence conditions under Gaussian data assumptions. It extends this framework to Linear Autoencoders (LAEs), introducing LAE-specific constraints, and proposes a computationally efficient evaluation method. Empirical validation on large-scale recommender datasets shows the bound is tight and correlates with practical metrics such as Recall and NDCG.

### Strengths

* **Novelty**: First PAC-Bayes bound for both multivariate regression and LAEs, addressing a theoretical gap.
* **Technical Soundness**: Rigorous and technically sound analysis, including convergence conditions.
* **Practical Relevance**: Efficient computational method.
* **Empirics**: Empirical results show tightness and correlation with performance.

### Weaknesses and missing elements

* Practical computation restricted by Gaussian/full-rank assumptions, limiting applicability.
* Discrepancy between theoretical squared-loss bound and ranking-based evaluation metrics.
* Some presentation and clarity issues (notation reuse, dense exposition, formatting).
* Initial literature review was incomplete (e.g., earlier PAC-Bayes results for regression and VAEs).

### Key reasons for acceptance

1. Novel and technically rigorous contribution filling an acknowledged gap in PAC-Bayes theory.
2. Relevance and applicability to recommender systems, with practical evaluation.
3. Rebuttal and discussion stage convincingly addressed literature positioning, clarity, and empirical concerns.

### Summary of discussion and rebuttal

Before rebuttal:

* `npXJ`: Criticized lack of references to prior PAC-Bayes work on multivariate regression and VAEs; recommended *Reject*.
* `7BK6`: Positive on theory and empirical validation but raised concerns about Gaussian assumptions and mismatch with ranking metrics.
* `dPHG`: Positive but questioned assumptions on input data and potential use of the bound in training.
* `fRvV`: Raised clarity and novelty positioning concerns; found paper difficult to parse.
* `z6h4`: Strongly positive; main concern was formatting/structure.

### Authors’ rebuttal

* Clarified distinctions between their bound and Mai (2022, 2023) on excess risk, and between LAEs and VAEs.
* Acknowledged limitations of Gaussian/full-rank assumptions, suggesting possible low-rank extensions.
* Justified squared-loss use as a surrogate, citing literature and empirical correlation with ranking metrics.
* Clarified notation, assumptions, and motivation for key lemmas and theorems.
* Agreed to improve literature review and presentation.

### Post-rebuttal positions

* `npXJ`: Upgraded to positive, calling the contribution “reasonably good” and suitable for NeurIPS.
* `7BK6`: Maintained borderline accept, satisfied with clarification but noting scope limits.
* `dPHG`: Maintained positive, satisfied with responses.
* `fRvV`: Updated to positive, stating changes requested would secure acceptance.
* `z6h4`: Maintained positive, satisfied with rebuttal.

### Weighing these points

All reviewers converged toward acceptance after rebuttal, with initial concerns on literature positioning, clarity, and evaluation satisfactorily addressed. While some limitations remain, these are well-documented, acknowledged by authors, and do not undermine the core contributions.